# Does credit rating agency reputation matter in China's local government bond market?

Changqian Xie, Rubi Ahmad, Eric H. Y. Koh *

Department of Finance, Faculty of Business and Economics, Universiti Malaya, Kuala Lumpur, Malaysia

* erickoh@um.edu.my

## Abstract

All issuers in China's local government bond market, which is nascent but growing rapidly, have the same AAA ratings. However, we provide evidence that the credit rating agency's reputation can certify differences in ratings' reliability and further impact bond pricing. On the basis of a sample of 7941 local government bonds issued from 2015 to 2021, results show that risk premium is significantly low for bonds rated by prestigious credit rating agencies, which means that issuers can save borrowing costs. Moreover, local governments regarded as less transparent in fiscal information disclosure enjoy more cost savings for their bonds by hiring more reputable agencies. Regression results are affirmed with the Heckman two-stage model, difference-in-differences regression, and machine learning method to solve the potential endogeneity issue. This paper's findings contribute to the debate on the credit rating agency's reputation hypothesis and present three implications. First, investors can rely on the credit rating agency's reputation to complement credit risk analysis. Second, local government policymakers should implement appropriate policies to reduce debt costs and improve public finance sustainability. Lastly, regulators should considerably focus on the supervision of credit rating agencies, given their substantial impact on bond pricing and the market's information asymmetry.

## Introduction

Local governments in developed and emerging countries play an important role in supporting the sustainable development of regional economies [1], including maintaining employment and providing healthcare during the COVID-19 pandemic [2, 3]. Local government bonds serve as essential sources of funds for these governments [4], accompanied by an increased scale worldwide in the past decades [5]. Scholars, practitioners, and policymakers have regarded the effective management of local government debt as a key driver of economic growth and sustainability [6]. Given that bond pricing is important in debt management, an enhanced understanding of the determinants of local government bonds' pricing can improve market efficiency and sustain public finance in the long term.

In the local government debt research, China presents an interesting and important case. According to data from the Wind database, the outstanding amount of China's local government bonds had increased from RMB200 billion ($31.37 billion, RMB to US dollars at an

information. If readers want to use the original data, they can visit the website: https://www.wind.com.cn/NewSite/edb.html.

**Funding:** 1) Recipient of this funding: Eric H.Y. Koh 2) Grant number: GA013-2021 3) Full name of funder: Tun Ismail Ali Chair 4) URL of funder website: https://tiac.um.edu.my/index 5) The funder had no role in study design, data collection and analysis, decision to publish, or preparation of the manuscript.

**Competing interests:** The authors have declared that no competing interests exist.

exchange rate of 6.3757:1 on December 31, 2021) at the end of 2009 to RMB30.30 trillion ($4.75 trillion) at the end of 2021. It has been the world's second-largest local government bond market. In addition, China's capital market has a strong underlying macroeconomic foundation and an increasingly open regulatory environment, which makes it an attractive investment target for international investors [7]. For example, from April 2019, the Bloomberg Barclays Global Aggregate Index included China's domestic sovereign and policy bank bonds. In April 2020, China lifted restrictions on foreign ownership of securities and fund management companies, enabling them to set up wholly owned subsidiaries in Chinese mainland. In October 2021, two local government bond issuers, namely, Guangdong and Shenzhen, successfully issued offshore bonds. However, evaluating credit risk in China's bond market is challenging because of the high proportion of top ratings. Over 80% of China's non-financial corporate bonds' ratings are AA or higher [2]. Furthermore, all local government bonds receive the same AAA rating. Therefore, many critics have argued that the China's bond rating provides limited meaningful information [8]. This situation is not conducive to the sustainable development of this market.

Credit rating agencies, as information intermediaries, provide incremental information to the market and affect bond pricing [9, 10]. These agencies particularly play two types of informational roles, namely, information revelation and information certification, in bond markets [11]. Information revelation refers to the disclosure of information on issuers' default risks via credit rating agencies' rating services. Given that undifferentiated ratings exist in China's local government bond market, this information revelation role is minimal. Meanwhile, information certification refers to the extent to which rating agencies' reputation may help certify or add credibility to the reliability of ratings. Given this background, questioning whether or not credit rating agencies' reputation still matters is natural. Therefore, this paper aims to examine the effects of the credit rating agency reputation on risk premiums of China's local government bonds at issuance.

This paper finds the following results. A negative relationship exists between risk premium and credit rating agency reputation on the sample of 7941 local government bonds issued from 2015 to 2021. Risk premiums on bonds rated by credit rating agencies with high reputation are lower than those rated by agencies with low reputation. On average, local government bonds rated by reputable credit rating agencies save over 2% of costs compared with those rated by non-reputable agencies. Considering the large-scale China's local government bond market, this reduction effect on borrowing costs is considerable for the economy. Furthermore, the extent to which credit rating agency reputation reduces risk premium is considerably prominent in local governments with low fiscal transparency.

A series of endogeneity tests affirm these results. First, the Heckman two-stage model is used to correct potential selection bias. Second, an analysis based on the propensity-score matching (PSM) method and difference-in-differences (DID) regression is conducted using the opening up of China's rating industry as an exogenous shock to address the potential endogeneity issue. Third, this paper constructs a counterfactual set of a part of the sample based on the machine learning method to solve the interference of potential reverse causality. Lastly, two robustness tests are provided to enhance the reliability of the research results.

This paper contributes to the literature in three ways. First, this research provides new empirical evidence to debate the credit rating agency reputation hypothesis. Noise from the information revelation role of credit rating agencies can be removed because all issuers in China's local government bond market receive the same AAA ratings. Therefore, we can focus on the impact of the credit rating agency's reputation on borrowing costs. Second, China's bond market is complex and appears opaque to foreign investors despite its increasing liberalization in recent years [2]. To alleviate this phenomenon, we contribute to the under-researched

context of China's rapidly growing bond market, which is attracting more international participants. Our findings suggest that investors can infer extensive underlying information by studying the credit rating agency's reputation. The engagement of a prestigious credit rating agency helps local governments alleviate investors' information asymmetry concerns and enhance confidence in their credit rating reliability. Lastly, this study implies that the credit rating agency's information certification role can work independently of the revelation role, and its effect on the cost of debt is also impacted by issuers' information transparency level.

The remainder of this paper is structured as follows. Section 2 reviews the pertinent literature on institutional background and past studies before developing the hypotheses. Section 3 presents the data and methodology. Section 4 reports the empirical results and robustness tests with analysis and discussion. Lastly, Section 5 provides the conclusion.

## Literature review

### Institutional background

China's credit rating industry emerged in the mid-1980s, coinciding with the development of the corporate bond market. The U.S. initially established credit rating agencies to meet investors' demand for information. By contrast, China's credit rating agencies were established because of the regulator's demands [7]. In the 1990s, the Chinese regulator mandated bond ratings prior to being publicly offered [11]. The Chinese rating industry structure, which is a non-monopoly industry with over 10 credit rating agencies, is different from that in the U.S. [7]. Before March 2018, the Chinese government prohibited foreign credit rating agencies from directly rating domestic Chinese bonds to protect the domestic credit rating industry [11]. Therefore, only in January 2019 and May 2020 did Standard & Poor (S&P) and Fitch become the first and second foreign credit rating agencies, respectively, to rate debt instruments in China's domestic bond market.

Although three big foreign credit rating agencies, that is, Moody's, S&P, and Fitch, were not allowed to rate directly before 2018, they were allowed to invest in domestic credit rating agencies with a regulatory cap of ownership. Specifically, in 2006, Moody's became a partner of CCXI Credit Rating Co. Ltd. with 49% ownership. In 2007, Fitch acquired 49% ownership of China Lianhe Credit Rating Co. Ltd. In 2008, S&P signed a technical cooperation agreement with Shanghai Brilliance Credit Rating & Investors Service Co. Ltd. This agreement was related to cooperation in many areas, such as training of credit rating personnel, cooperation in credit rating research, and sharing of credit rating technology. At present, the local government bond market has seven credit rating agencies, as shown in Table 1, including three cooperating with global partners.

**Table 1. List of credit rating agencies.** This table presents the basic information about credit rating agencies in China's local government bond market.

| Name of credit rating agency | Global partner | Website |
|---|---|---|
| CCXI Credit Rating Co. Ltd. (CCXI_Moody) | Moody's | http://www.ccxi.com.cn/ |
| China Lianhe Credit Rating Co. Ltd. (Lianhe_Fitch) | Fitch | http://www.lhratings.com/ |
| Shanghai Brilliance Credit Rating & Investors Service Co. Ltd. (Brilliance_S&P) | S&P | http://www.shxsj.com/ |
| Dongfang Credit Rating Co. Ltd. (Dongfang) | No | https://www.dfratings.com/ |
| CSCI Pengyuan Co. Ltd. (CSCI) | No | https://www.cspengyuan.com/ |
| China Bond Rating Co. Ltd. (CBR) | No | https://www.chinaratings.com.cn/ |
| Dagong Global Credit Rating Co. Ltd. (Dagong) | No | https://www.dagongcredit.com/ |

The development of China's local government bond market has four stages: from the introduction stage in 2009 to the present.

In the first stage (2009–2010), the National People's Congress (NPC) designated the Ministry of Finance to act as agent for local governments. The Ministry of Finance was responsible for the issuance and repayment of local government bonds. That is, although local governments were the nominal issuers, they could only receive and repay funds via the Ministry of Finance. A total of 60 local government bonds (RMB40 billion) were issued in this stage.

In the second stage (2011–2013), the State Council allowed six local governments (Zhejiang, Guangdong, Jiangsu, Shandong, Shanghai, and Shenzhen) to issue their own bonds on a pilot basis, but the Ministry of Finance was still responsible for fulfilling the repayment obligation. A total of 25 local bonds (RMB80 billion) were issued in this stage.

In the third stage (2014), the Ministry of Finance published a document titled *Regulations for the Self Issuance and Self Repayment of Local Government Bonds in Pilot Areas*. Four local governments (Jiangxi, Ningxia, Beijing, and Qingdao) and the six local governments in the second stage were allowed to directly issue and repay bonds. All other local governments not included on the pilot list still had to issue and repay the local government bonds through the Ministry of Finance. A total of 43 local government bonds (RMB40 billion) were issued in 2014.

In the fourth stage (2015-present), China's central government lifted the restriction on autonomous bond issuance for all provinces, autonomous regions, centrally administered municipalities, and five Municipalities with Independent Planning Status (MIPS). The amended *Budget Law* indicated that the local government bond issuance is the only legal way for local governments to raise funds; they are prohibited from raising funds through bank loans or other financing channels. Table 2 presents an overview of bond issuance from 2015 to 2021. Compared with a relatively steady issuance in the first three stages, the issuance amount has experienced a rapid increase since 2015. The outstanding amount at the end of 2021 is about 6.3 times as many as that in 2015.

## Hypothesis development

**Information asymmetry and rating inflation.**   A fundamental problem with risky debt or equity issuance is that insiders or shareholders may take advantage of their non-public information advantages [12, 13]. Issuers (i.e., a type of insider) possess non-public information on their issuance security. Therefore, opportunistic insiders in an information asymmetry environment exploit informational advantages over outsiders to extract benefits.

For local government bond issuers, information advantages in the opaque market environment include but not limited to the following aspects. Detailed information on the

**Table 2. Local government bond issuance in China (2015–2021).** This table presents China's local government bond issuance from 2015 to 2021. Data are obtained from the Wind Database.

| Year | Issuance amount (RMB, billion) | Number of issuances | Outstanding amount at the end of year (RMB, billion) |
|------|--------------------------------|---------------------|------------------------------------------------------|
| 2015 | 3,835.06 | 1,035 | 4,826.01 |
| 2016 | 6,045.84 | 1,159 | 10,628.18 |
| 2017 | 4,358.09 | 1,134 | 14,744.82 |
| 2018 | 4,165.17 | 930 | 18,069.95 |
| 2019 | 4,362.43 | 1,093 | 21,118.29 |
| 2020 | 6,443.81 | 1,848 | 25,486.41 |
| 2021 | 7,482.63 | 1,991 | 30,299.58 |

performance of local governments is not disclosed to the public or taxpayers. This situation complicates the process of adequately assessing and monitoring governments' financial health [14]. Moreover, the local government bond market's infrequent trading and low default incidences indicate less new information on issuers than in the corporate bond and equity markets [15]. Lastly, local governments are not subject to the regulator's compliance requirements compared with corporations [16, 17].

To alleviate this information asymmetry, credit rating agencies are in a unique position to reveal information on the issuer's default risk and to certify the quality of risky securities [18]. However, following the 2008–2009 financial crisis, there has been increasing concerns with credit rating reliability, particularly rating inflation [18–20]. This strand of literature has identified at least four underlying causes of rating inflation.

First, the issuer-pay business model allows rating agencies to obtain some or all benefits by providing high ratings, thereby resulting in conflict of interest [21, 22]. Existing major rating agencies earn their income from issuers rather than investors, unlike the case when John Moody started his rating company in 1905 [19]. Second, regulations have played a vital role in the upward bias of ratings by linking capital and investment requirements to ratings [20, 21]. In computing regulatory capital requirements under Basel III, single A-rated municipal bonds are assigned risk weights of 20%, higher than Aa- or Aaa-rated municipal bonds that are assigned 0% risk weight. Commercial banks are major investors in this market. This regulation incentivizes them to invest in higher-rated bonds, regardless of whether or not ratings accurately reveal underlying risks. In particular, rating agencies can make "subjective" adjustments to their rating models, thereby catering to investors' demand for regulatory arbitrage [23]. Third, ratings will likely be inflated when issuers can hire several credit rating agencies and disclose only the most favorable ratings; this phenomenon is called "rate shopping behavior" [18, 20]. This competition among rating agencies results in minimal information revealed and creates additional shopping opportunities for issuers. Lastly, rating inflation is markedly common during booms [15] and when investors are naïve [24].

**Reputation certification hypothesis.** The rating inflation phenomenon indicates that information revealed by credit rating agencies may not be totally credible and reliable. In this case, clarifying the extent to which we can rely on ratings issued by different credit rating agencies is of practical significance. Accordingly, a potential solution is to focus on the credit rating agency's reputation. Given that reputation is an intangible asset that is valuable, rare, and difficult to imitate, firms with good reputations maintain a sustained competitive advantage [25].

In the realm of finance, the role of financial intermediaries' reputations, specifically of underwriters, has drawn extensive attention in alleviating information asymmetry. Booth and Smith [12] established the reputation certification hypothesis by demonstrating the role of underwriters' reputations in certifying risky securities' quality and pricing. Chemmanur and Fulghieri [25] developed a theoretical model that can be applied to numerous intermediaries, such as auditors, financial analysts, investment banks, and bond rating agencies. Their model demonstrated that applying strict evaluation standards is costly for information intermediaries in the short run but beneficial in the long run. The reason is that such an application rewards information intermediaries with good reputations.

**Naïve and rational investors assumption.** Most theoretical discussions on corporate finance literature have conjectured that reputation will generate positive outcomes for firms. However, differences among the assumptions on whether bond investors are naïve or rational have resulted in debates on the reputation certification effects of rating agencies [7].

Theoretical models of Bar-Isaac and Shapiro [15], Bolton, Freixas, and Shapiro [20], and Skreta and Veldkamp [24] have relied on investors' behavioral biases (i.e., presence of naïve investors). Bar-Isaac and Shapiro [15] suggested that credit rating agencies' reputational

concerns vary across business cycles and result in countercyclical rating qualities. Therefore, introducing naïve investors into the model worsens the overall rating quality but does not change the countercyclical feature of rating accuracy. Bolton, Freixas, and Shapiro [20] assumed that a higher proportion of trusting investors than sophisticated investors contributes to the emergence of less reliable ratings. Skreta and Veldkamp [24] further asserted that investors cannot rationally adjust for upward bias in reported ratings owing to financial products' increasing complexity. On the basis of this strand of theoretical literature, the effect of rating agencies' reputation certification is reduced by investors' low ability to properly identify rating quality.

Rather than relying on naïve investors, the assumption that investors are rational implies that they are not misled by existing rating inflation in equilibrium [21]. Several studies have used the rational investor hypothesis as basis, in suggesting that investors can differentiate among credit rating agencies and require higher risk premiums for bond issues subject to greater risk of an upward biased rating [18, 21]. Consistent with rational investor behavior, Sangiorgi and Spatt [18] found that investors require a higher yield for a new bond if an agency's rating of the issuer's bonds was one notch higher on average than other agencies' ratings in the previous year. Investors' rationality guarantees high-quality ratings in the market. Moreover, issuers are penalized for borrowing costs when they employ rating agencies with poor reputations; this situation is called the "winner's curse".

In the specific view of China's local government bond market, preliminary analysis of the available data indicated that in a situation where all other issuance characteristics are highly similar except the reputation of the rating agencies employed by issuers, there are still evident differences among risk premiums of each local government bond in China. For example, Heilongjiang and Shaanxi issued bonds (bond codes: 157774.IB and 104641.IB, respectively) on 24th June 2019. Most of the main issuance characteristics of the two bonds are highly similar, including the same maturity (30 years), bond type (general obligation bonds), sale method (public offering), and issue sizes (RMB10.24 billion and RMB11.06 billion, respectively). However, risk premiums for Heilongjiang and Shaanxi are 32.93 basis point (bp) and 24.93 bp, respectively. In this case, the naïve investor assumption fails to explain this phenomenon. That is, investors do not react myopically to ratings in China's bond market. They likely seek information from the underlying reputations of credit rating agencies and other sources to complement their analyses.

Moreover, commercial banks and other institutional investors are the main investors in China's local government bond market. According to data from the report of China's Bond Market Overview 2021, commercial banks were the largest holders of local government bonds by the end of 2021, accounting for 83.60% of the local government bond market. Other institutional investors, such as credit cooperatives, insurance companies, securities firms, unincorporated products, and overseas institutions, held 8.56% of local government bonds at the end of 2021. That is, individual investors held only 7.84% of local government bonds. Compared with individual investors, institutional investors tend to be more rational and often make investment decisions based on comprehensive information [26, 27].

Furthermore, China's central government has committed to guiding the establishment of a market-oriented local government bond market and stating it clearly that the central government follows the no bailout principle. In 2014 and 2016, the State Council issued two documents titled *Opinions of the State Council on Strengthening the Management of Local Government Debts* and *Emergency Response Plan for Local Government Debt Risks*. Both documents clearly state that local governments are the only subjects responsible for repaying their debts. Therefore, this paper should sensibly incorporate the rational investor assumption into the analysis.

We use the rational investor assumption and reputation certification theory as bases in inferring that credit rating agencies' reputational differences lead to yield differentials. The first hypothesis is stated as follows:

**Hypothesis 1 (H1):** *Employing credit rating agencies with high reputation helps local governments reduce their bonds' risk premiums.*

**Transparency and reputation certification effect.**    If *H1* is true, then we are further interested in how issuers' self-disclosure influences the effect of credit rating agencies' reputation certification. That is, we aim to examine the moderating effect of issuers' self-disclosure on the relationship between credit rating agency's reputation and local government bond's risk premium. Issuers' self-disclosure indicates that they report information on local governments' public finance revenue, expenditure, and economic performance. If issuers' self-disclosure contents are broader and of higher quality, they are generally considered to have higher fiscal transparency by market participants. Previous literature has demonstrated that issuers' high fiscal transparency leads to lower financing costs because the uncertainty faced by investors in risk analysis is reduced [28–30]. Thus, this study will no longer repeatedly discuss the impact of issuers' disclosure transparency on risk premiums.

A recent study [31] linked issuers' information disclosure and credit rating agencies from the information disclosure perspective groundbreakingly. When managers have autonomy in information disclosure, their different degrees of voluntary disclosures on firm information affect credit ratings apart from borrowing costs. Similar to firms' managers, local government officials also have a certain degree of autonomy in information disclosure, leading to differences in local governments' fiscal transparency [32, 33]. In particular, there are two main underlying reasons for this difference in the fiscal transparency of local government bond markets. On the one hand, this discretion over information disclosure enables local government officials to pursue their political goals. That is, local government officials face a trade-off between political and borrowing costs in the decision on providing transparency. Different political goals of each local government official make the statuses of fiscal transparency differ among each local government. On the other hand, local governments differ in resource endowment, accessibility to public finance revenues, and fiscal management ability, leading to heterogeneity in the quantity and quality of financial information disclosed.

Although the difference in issuers' fiscal transparency has not resulted in a difference in ratings of China's local government bonds in our study, there is a possibility of influencing the aforementioned effect of credit rating agencies' reputation certification on risk premiums. In developing *H1*, it is clarified that rational investors will analyze the credit rating agencies' reputation to obtain supplementary information in the market with information asymmetry. Given that issuers' self-disclosure can also mitigate the opacity of market information, the roles of issuers' self-disclosure and rating agencies' reputation will likely complement each other. Therefore, investors will rely less on the obtained supplementary information from the credit rating agency's reputation if they can acquire additional information from the issuers' self-disclosure. In this case, the importance of the same reputation of credit rating agencies become different for investors in environments with different information asymmetry. That is, the extent to which investors depend on the credit rating agency's reputation is influenced by the degree of issuers' self-disclosure. The higher the degree of the issuers' self-disclosure means they are confident and willing to report information to the public, thereby suggesting that their hidden risk is low, and vice versa. This situation is similar to the viewpoint in previous literature that investors tend to equate no news with bad news when they analyze firms [31, 34–36].

We use the preceding analysis as basis to conjecture that with an increase in information disclosure of local government bond issuers (i.e., increase in fiscal transparency), investors will

pay less attention to the credit rating agencies' reputation. By contrast, local governments with low fiscal transparency can enjoy additional reduction in borrowing costs by employing credit rating agencies with high reputation. Hence, the following hypothesis will be tested:

*Hypothesis 2 (H2): Credit rating agency's reputation effect is more important for local governments with lower fiscal transparency to reduce risk premiums.*

## Data and methods

### Sample selection

China's local government bond market has 37 issuers, including 31 mainland provincial-level governments, five Municipalities with Independent Planning Status (MIPS) governments, and one Xinjiang Production and Construction Corps (Corps) government. We exclude the latter six local governments because they do not have the same rights as the 31 mainland provincial-level governments in administrative and management aspects. We also exclude option-embedded bond issues because of their complex and unique pricing techniques.

The final sample consists of 7941 bond issue observations issued by 31 mainland provincial-level governments from 2015 to 2021. Two reasons are cited for the period starting with 2015. First, China's local government bond market was very small before 2015 but experienced significant growth thereafter. Second, most local government bonds before 2015 were guaranteed by the central government. Hence, these bonds do not fully represent local governments' credit risk. Data on the basic information of bond issues are collected from the Wind database.

Table 3 presents the sample breakdown by (1) bond type, (2) maturity, (3) issuance year, (4) credit rating agency, and (5) region. For bond maturity (Panel B), local government bonds with four maturity periods (i.e., 3 years, 5 years, 7 years, and 10 years) account for 80.22% of the entire sample. This result is consistent with the fact that the four maturity types are key bond maturities assigned by the Ministry of Finance. Only 1571 local government bonds (about 19.78% of the sample) have the other five maturity periods (i.e., 1 year, 2 years, 15 years, 20 years, and 30 years). In Panel C, the mean risk premium shows its peak value in 2018 because of the strict regulations in China's bond market that started in 2017, and decreased gradually thereafter. Evidently, the number of bonds issued in 2020 and 2021 increased substantially.

Panels D and E of Table 3 show the distributions related to credit rating agencies and regions, respectively. A total of 37.55% of 7941 local government bonds are rated by credit rating agencies with global partners. The mean risk premium of local government bonds rated by CCXI_Moody is the lowest. CBR, a pure local credit rating agency, takes up the largest market share of ratings in China's local government bond market. The 31 mainland provinces are divided into four regions (i.e., east, central, west, and northeast) according to geographical location. In detail, the east region includes ten issuers (Beijing, Fujian, Guangdong, Hainan, Hebei, Jiangsu, Shandong, Shanghai, Tianjin, and Zhejiang); the central region includes six issuers (Anhui, Henan, Hubei, Hunan, Jiangxi, and Shanxi); the west region includes 12 issuers (Chongqing, Gansu, Guangxi, Guizhou, Inner Mongolia, Ningxia, Qinghai, Shaanxi, Sichuan, Tibet, Xinjiang, and Yunnan); the northeast region includes three issuers (Jilin, Liaoning, and Heilongjiang). The mean risk premiums of local government bonds issued by provinces in the west and northeast regions are higher than those issued by provinces in the east and central regions.

### Research method

Given that the dependent variable (*Risk Premium*) is measured at the bond level rather than the local government level, this paper employs pooled data and multivariate ordinary least

**Table 3. Distribution of sample.** This table reports the distribution of the sample bond observations by bond type, maturity, year, credit rating agency, and region. The percentage (%) is calculated as the number of bond issuances assigned to each breakdown category divided by the total number of 7941 observations.

| | Number of bonds | Percentage (%) | Mean risk premium (bp) |
|---|---|---|---|
| **Panel A: By bond type** | | | |
| General obligation bonds | 2884 | 36.32 | 28.72 |
| Revenue bonds | 5057 | 63.68 | 28.31 |
| **Panel B: By maturity** | | | |
| 1 year | 13 | 0.16 | 25.53 |
| 2 years | 28 | 0.35 | 26.79 |
| 3 years | 938 | 11.81 | 26.39 |
| 5 years | 1950 | 24.56 | 30.43 |
| 7 years | 1510 | 19.02 | 29.48 |
| 10 years | 1972 | 24.83 | 29.69 |
| 15 years | 613 | 7.72 | 24.63 |
| 20 years | 490 | 6.17 | 24.99 |
| 30 years | 427 | 5.38 | 24.29 |
| **Panel C: By year** | | | |
| 2015 | 924 | 11.64 | 26.99 |
| 2016 | 1046 | 13.17 | 22.10 |
| 2017 | 1055 | 13.29 | 37.03 |
| 2018 | 849 | 10.69 | 43.97 |
| 2019 | 972 | 12.24 | 27.31 |
| 2020 | 1553 | 19.56 | 25.42 |
| 2021 | 1542 | 19.42 | 23.00 |
| **Panel D: By credit rating agency** | | | |
| CCXI_Moody | 422 | 5.31 | 21.20 |
| Lianhe_Fitch | 718 | 9.04 | 29.68 |
| Brilliance_S&P | 1842 | 23.20 | 29.30 |
| Dongfang | 818 | 10.30 | 28.17 |
| CSCI | 205 | 2.58 | 25.26 |
| CBR | 3347 | 42.15 | 28.10 |
| Dagong | 589 | 7.42 | 33.00 |
| **Panel E: By region** | | | |
| East | 2672 | 33.65 | 26.38 |
| Central | 1744 | 21.96 | 26.83 |
| Northeast | 614 | 7.73 | 32.24 |
| West | 2911 | 36.66 | 30.53 |
| **Total** | 7941 | 100.00 | 28.45 |

squares (OLS) regression to test the proposed research hypotheses. The baseline regression model is presented as Eq (1).

$$Risk\ Premium_{i,p,t} = \beta_0 + \beta_1 Reputation\_CRA_{i,p,t} + \beta_2 Fiscal\ Transparency_{i,p,t-1} +$$

$$\beta_3 RepFis_{i,p,t} + Control\ Variables + Year\ Dummies + Region\ Dummies + \qquad (1)$$

$$Issuer\ Dummies + \varepsilon_{i,p,t}$$

where the subscripts $i$, $p$, and $t$ represent the bond, issuer, and year, respectively.

The regression model includes the issuer dummy variables to control for potential bias owing to omitted variables and year dummy variables to control for potential temporal effect. In addition, region dummy variables are considered in the baseline regression model. The regression model is also adjusted for robust standard error to control for biased standard error. We winsorize the top $1^{st}$ and bottom $99^{th}$ percentiles of the continuous variables to control outliers bias. For control variables of the issuer characteristics, we use the data in the previous year ($t$-1), to relatively mitigate endogeneity problems [26].

To further validate the regression model, this paper employs the Heckman two-stage model, DID OLS regression, and machine learning method to address potential endogeneity issues. The details of each method will be introduced in the corresponding sections.

## Variable description

**Risk premium.** We adopt a widely used approach,and measure risk premium (*Risk Premium 0*) as the difference between the local government bond yield at issuance date versus that of the Chinese central government bond of the same maturity [37, 38]. We also adopt a measure often used in Chinese studies because of its conventional market rule. The second measure (*Risk Premium 5*) calculates the difference between the same issuance yield versus the central government bond average yield of the past five working days [39]. For robustness, baseline regressions of both measurements of risk premium are conducted.

**Credit rating agency reputation.** Previously, market share [40] and third-party ranking [41] were the most common proxies to measure credit rating agency's reputation. Some have argued that market share does not indicate Chinese credit rating agency reputation because it represents the degree of competition in the rating industry rather than reputation [42]. Third-party ranking by the National Development and Reform Commission (NDRC) covers only six credit rating agencies that provide enterprise bond ratings. CBR, which provides rating services for 13 local governments, is not covered by NDRC's ranking list. Therefore, both proxies are not appropriate for measuring rating agencies' reputation in this study. In addition, some researchers [43, 44] have used bankruptcy of companies with previous high ratings as negative shocks to study the reputation of rating agencies. However, no default event happened in China's local government bond market until now, resulting in this method being unsuitable for our study as well.

Some recent findings have suggested that the different types of credit rating agencies' ownership are appropriate proxies to measure the reputation of credit rating agencies in China [7, 11, 45, 46]. In the Japanese bond market, ratings issued by more reputed global credit rating agencies have a stronger effect than those issued by local ones [47]. On the basis of this criterion, credit rating agencies with foreign ownership are recognized as having high reputation, while the rest have low reputation. Therefore, CCXI_Moody, Lianhe_Fitch, and Brilliance_S&P are in the high-reputation group. Based on the above analysis, the coefficient of credit rating agency reputation variable (*Reputation_CRA*) is expected to be significantly negative.

**Local government's fiscal transparency.** To test *H2*, two different variables are included in the regression: (1) variable of local government's fiscal transparency (*Fiscal Transparency*) and (2) interaction term (*RepFis*) of fiscal transparency and credit rating agencies' reputation variable. *Fiscal Transparency* is measured as the provincial governments' fiscal transparency index published by Shanghai University of Finance and Economics, which is a widely used indicator in studying China's provincial fiscal transparency [48]. This indicator includes an assessment of provincial governments' voluntary disclosure of information in several aspects, such as public finance budget, operation performance of local government-owned enterprises, and specific fiscal accounts. To date, Shanghai University of Finance and Economics has

released data on the 31 provinces' fiscal transparency for 10 years (2009 to 2018). Note that the data will no longer be updated because the organization believes that fiscal transparency level of each subject has plateaued. In the regression, we use data with a lag of one year. That is, fiscal transparency data from 2014 to 2018 are used to match the dependent variable from 2015 to 2019. Meanwhile, we keep the data of fiscal transparency for 2020 and 2021 the same as in 2019. It is worth noting that, when the fiscal transparency data with a two-year lag is used to replace the one-year-lag data, the results are similar. For the *RepFis*, we are interested in its coefficient, which is expected to be positive. This result suggests that the reputational effect of credit rating agencies is markedly important for reducing risk premiums for local governments with low fiscal transparency.

**Control variables.** Other regressors, including issue-specific, issuer-specific, and other variables, are treated as control variables in our study. Full definitions of the variables are shown in Table 4.

Issue-specific variables include bond features that are significant determinants of offering yields. We refer to the previous literature and include *Maturity*, *Issue Size*, *Issue Frequency*, *Bond Type*, and *Sale Methods* to control for issue-specific characteristics. *Maturity* is expected to be positively related to risk premium [17]. Size effect (*Issue Size*) may have either a positive

**Table 4. Variable definition.** This table presents definitions for the variables studied.

| Variable Name | Definition |
|---|---|
| Risk Premium 5 | Difference (in bp) between the local government bond's yield at issuance and the average yield of China's central government bond with the same maturity during the past five working days. |
| Risk Premium 0 | Difference (in bp) between the local government bond's yield at issuance and the China's central government bond's yield with the same maturity. |
| Reputation_CRA | A dummy variable, which equals one if the bond is rated by CCXI_Moody, Lianhe_Fitch, or Brilliance_S&P, and zero otherwise. |
| Fiscal Transparency | The natural logarithm of the fiscal transparency index for each issuer in the previous year. |
| RepFis | The interaction term of fiscal transparency and credit rating agencies' reputation variable (Reputation_CRA * Fiscal Transparency). |
| **Issue-specific Variables** | |
| Maturity | Number of years to the maturity of a particular bond issue at the time of issuance. |
| Issue Size | The natural logarithm of one plus the face value of a particular bond (in million, RMB) at the time of issuance. |
| Issue Frequency | The natural logarithm of the number of total issuance times in the same year for each issuer. |
| Bond Type | A dummy variable, which equals one if a particular bond issue is a revenue bond, and zero otherwise. |
| Sale Methods | A dummy variable, which equals one if a particular bond issue is issued by private placement, and zero otherwise. |
| **Issuer-specific Variables** | |
| GDP Year | The natural logarithm of the issuer's total GDP (in 100 million, RMB) in the previous year. |
| GDP per capita | The natural logarithm of the GDP per capita (in yuan, RMB) of the issuer in the previous year. |
| GDP Growth Rate | The GDP growth rate (%) of the issuer in the previous year. |
| Fixed Asset Investment (FAI) Growth Rate | The fixed asset investment growth rate (%) of the issuer in the previous year. |
| Debt Ratio | The ratio of the outstanding amount of local government debt to the issuer's comprehensive fiscal revenue (%) in the issuance year [1]. |
| Public Revenue per capita | The natural logarithm of the public revenue per capita (in yuan, RMB) of the issuer in the previous year. |
| Public Revenue Growth Rate | The issuer's public revenue growth rate (%) in the previous year. |
| **Other Variables** | |
| Tbill | The interest rate (%) of the one-year central government bond on the date of the local government bond's issuance. |

[1] The data on the outstanding amount of local government debt has been published since 2015, and the data in 2014 is unavailable (http://www.celma.org.cn/). In order to reduce the sample loss, we measure the variable of Debt Ratio based on the year of bond issuance. When we reduce the sample period and use the data on *Debt Ratio* in the previous year, the results are similar to the current baseline results.

[11]or negative [49] influence on risk premiums. *Issue Frequency* is expected to be negative with risk premiums. The reason for this expectation is that when one issuer frequently enters the bond market, its degree of information disclosure accumulates gradually, resulting in low risk premium [50]. *Bond Type* is also included as a control variable. The two types of local government bonds are general obligation and revenue bonds. General obligation bonds are those backed with the "full faith and credit" of local governments. Therefore, this type of bond is charged a low risk premium [17]. *Sale Methods*, such as private placement and public offering sale method, also influence bonds' risk premiums [51].

Issuer-specific variables refer to the issuer characteristics. This study includes the effect of fiscal performance, which is found to significantly impact a local government's default risk [52]. Higher debt ratio and lower fiscal revenue imply a higher default risk, in turn resulting in higher risk premium. *Debt Ratio* is positively related to risk premium [37]. The fixed asset investment growth rate of the issuer in previous year (*FAI Growth Rate*) is also included. Higher level of *FAI Growth Rate* tends to be accompanied by a higher debt burden, which is expected to be positively related to risk premium. Fiscal revenue ability (*Public Revenue Growth Rate* and *Public Revenue per capita*) is expected to be positively related to risk premium [39, 51]. We also include the issuer's previous year GDP performances, such as *GDP Growth Rate*, *GDP per capita*, and *GDP Year*. All GDP performances are expected to be negatively related to risk premium [39, 51, 53].

Lastly, *Tbill* is set as another control variable to determine time movements in the bond market.

## Summary statistics

Tables 5 and 6 present the summary statistics and pairwise correlation analysis of the variables, respectively.

For summary statistics, the mean, median, and standard deviation of the dependent variable measured by *Risk Premium 0* and *Risk Premium 5* are fairly similar. The minimum of *Risk*

**Table 5. Summary statistics of variables.** This table reports the summary statistics of the variables included in the analysis for a sample of 31 local governments.

| Variable | Obs | Mean | Median | Standard deviation | Min | Max |
|---|---|---|---|---|---|---|
| Risk Premium 5 | 7941 | 28.6754 | 25.5700 | 14.2383 | -11.6180 | 85.7800 |
| Risk Premium 0 | 7941 | 28.4539 | 25.1760 | 14.1781 | -16.7400 | 89.8200 |
| Reputation_CRA | 7941 | 0.3755 | 0.0000 | 0.4843 | 0.0000 | 1.0000 |
| Fiscal Transparency | 7941 | 3.8151 | 3.9160 | 0.3616 | 2.7318 | 4.2486 |
| RepFis | 7941 | 1.4279 | 0.0000 | 1.8544 | 0.0000 | 4.2486 |
| Maturity | 7941 | 9.4104 | 7.0000 | 6.5775 | 1.0000 | 30.0000 |
| Issue Size | 7941 | 7.6607 | 7.8095 | 1.3638 | 0.6419 | 10.9495 |
| Issue Frequency | 7941 | 3.7775 | 3.7136 | 0.5636 | 1.6094 | 5.2781 |
| Bond Type | 7941 | 0.6368 | 1.0000 | 0.4809 | 0.0000 | 1.0000 |
| Sale Methods | 7941 | 0.1390 | 0.0000 | 0.3460 | 0.0000 | 1.0000 |
| GDP Year | 7941 | 10.0461 | 10.1271 | 0.8795 | 6.9499 | 11.6187 |
| GDP per capita | 7941 | 10.9507 | 10.9230 | 0.4314 | 7.6089 | 12.0130 |
| GDP Growth Rate | 7941 | 7.2908 | 7.8313 | 3.9709 | -5.3369 | 21.2441 |
| FAI Growth Rate | 7941 | 6.2211 | 7.6000 | 9.1813 | -56.6000 | 23.4000 |
| Debt Ratio | 7941 | 168.8497 | 153.5076 | 89.4300 | 28.0193 | 527.8049 |
| Public Revenue per capita | 7941 | 8.7242 | 8.5860 | 0.5084 | 7.8853 | 10.2709 |
| Public Revenue Growth Rate | 7941 | 4.2009 | 4.4400 | 7.8319 | -33.3700 | 24.0000 |
| Tbill | 7941 | 2.5098 | 2.4091 | 0.5069 | 1.1177 | 3.7979 |

**Table 6. Correlation matrix.** This table presents the correlation matrix of the variables.

| | Risk Premium 5 | Risk Premium 0 | Reputation_CRA | Fiscal Transparency | Maturity | Issue Size | Issue Frequency | Bond Type | Sale Methods |
|---|---|---|---|---|---|---|---|---|---|
| Risk Premium 5 | 1.0000 | | | | | | | | |
| Risk Premium 0 | 0.9651 | 1.0000 | | | | | | | |
| Reputation_CRA | -0.0152 | -0.0112 | 1.0000 | | | | | | |
| Fiscal Transparency | -0.0575 | -0.0561 | -0.0272 | 1.0000 | | | | | |
| Maturity | -0.0991 | -0.0999 | 0.0056 | 0.2637 | 1.0000 | | | | |
| Issue Size | -0.0711 | -0.0679 | 0.0521 | -0.1390 | -0.0314 | 1.0000 | | | |
| Issue Frequency | -0.1122 | -0.1154 | 0.0686 | 0.3813 | 0.2422 | -0.1980 | 1.0000 | | |
| Bond Type | -0.0075 | -0.0137 | 0.0243 | 0.2091 | 0.2086 | -0.3073 | 0.2315 | 1.0000 | |
| Sale Methods | 0.5267 | 0.5210 | -0.0057 | -0.2674 | -0.1867 | -0.0277 | -0.1499 | -0.1711 | 1.0000 |
| GDP Year | -0.1253 | -0.1166 | 0.1007 | 0.2745 | 0.1138 | 0.1891 | 0.4695 | 0.1552 | -0.1253 |
| GDP per capita | -0.1038 | -0.0960 | 0.0306 | 0.2098 | 0.0927 | 0.0526 | 0.1985 | 0.1285 | -0.1388 |
| GDP Growth Rate | 0.1873 | 0.2107 | -0.1215 | -0.1679 | -0.1128 | 0.0381 | -0.2543 | -0.0330 | 0.0543 |
| FAI Growth Rate | -0.0261 | -0.0151 | -0.0256 | -0.2683 | -0.0919 | 0.0676 | -0.0536 | -0.0651 | 0.0998 |
| Debt Ratio | 0.0985 | 0.0916 | 0.0138 | -0.1657 | 0.0994 | -0.1222 | -0.0545 | -0.0223 | -0.0129 |
| Public Revenue per capita | -0.1168 | -0.1116 | 0.0965 | 0.0987 | 0.0072 | 0.0543 | -0.0119 | 0.0666 | -0.0347 |
| Public Revenue Growth Rate | -0.0396 | -0.0316 | -0.1028 | -0.2157 | -0.1148 | 0.0860 | -0.2429 | -0.0945 | 0.1335 |
| Tbill | 0.3947 | 0.3830 | -0.0015 | -0.0712 | -0.2763 | -0.0061 | -0.2206 | -0.0494 | 0.0672 |
| | GDP Year | GDP per capita | GDP Growth Rate | FAI Growth Rate | Debt Ratio | Public Revenue per capita | Public Revenue Growth Rate | Tbill | |
| GDP Year | 1.0000 | | | | | | | | |
| GDP per capita | 0.4178 | 1.0000 | | | | | | | |
| GDP Growth Rate | -0.0544 | -0.1543 | 1.0000 | | | | | | |
| FAI Growth Rate | 0.0711 | -0.1689 | 0.4063 | 1.0000 | | | | | |
| Debt Ratio | -0.6019 | -0.3498 | -0.0976 | -0.1348 | 1.0000 | | | | |
| Public Revenue per capita | 0.1911 | 0.6650 | -0.0658 | -0.1174 | -0.3977 | 1.0000 | | | |
| Public Revenue Growth Rate | 0.0262 | -0.1719 | 0.5375 | 0.4583 | -0.2622 | 0.0482 | 1.0000 | | |
| Tbill | -0.0612 | -0.0719 | 0.2194 | -0.0365 | -0.0801 | -0.0690 | -0.0357 | 1.0000 | |

*Premium 5* is -11.6180 bp from a bond issued by Guangdong Province. The negative value of the minimum risk premium implies the existence of political intervention in issuance process from local governments with a strong economy. The average of the main independent variable, *Reputation_CRA*, represents proportion of the bonds rated by prestigious credit rating agencies to the overall observations. Given that the value of *Fiscal Transparency* is in the natural logarithm format, the average of *Fiscal Transparency* (3.8151) means that the original average level of fiscal transparency is 45.38 in 100. This result indicates that China's local government bond market has a relatively high degree of information asymmetry. The situation is similar to the local government bond markets in the U.S. and France [14, 16, 17].

For the pairwise correlation analysis, values of most coefficients are below 0.5. Thus, the correlation matrix does not suggest any serious multicollinearity concerns. Furthermore, multicollinearity issue was cross-verified through the variance inflation factor (VIF), in which the mean VIF for all explanatory variables in baseline regression is 3.29.

To avoid the potential issue that the core explanatory variable (*Reputation_CRA*) determines the differences between regions rather than those between the high and low reputations

**Table 7. Number of bonds rated by the credit rating agencies with high or low reputation in each region each year.**

| Year | Reputation of credit rating agencies | Regions | | | |
|---|---|---|---|---|---|
| | | East | Central | West | Northeast |
| 2015 | *Reputation_CRA=1* | 135 | 36 | 88 | 30 |
| | *Reputation_CRA=0* | 223 | 109 | 248 | 55 |
| 2016 | *Reputation_CRA=1* | 172 | 72 | 95 | 21 |
| | *Reputation_CRA=0* | 193 | 99 | 337 | 57 |
| 2017 | *Reputation_CRA=1* | 182 | 89 | 98 | 19 |
| | *Reputation_CRA=0* | 168 | 113 | 321 | 65 |
| 2018 | *Reputation_CRA=1* | 173 | 42 | 94 | 48 |
| | *Reputation_CRA=0* | 95 | 109 | 258 | 30 |
| 2019 | *Reputation_CRA=1* | 195 | 27 | 78 | 50 |
| | *Reputation_CRA=0* | 115 | 192 | 297 | 18 |
| 2020 | *Reputation_CRA=1* | 303 | 36 | 131 | 95 |
| | *Reputation_CRA=0* | 196 | 341 | 417 | 34 |
| 2021 | *Reputation_CRA=1* | 279 | 254 | 80 | 60 |
| | *Reputation_CRA=0* | 243 | 225 | 369 | 32 |

of credit rating agencies, Table 7 provides details of the number of bonds rated by the credit rating agencies with high or low reputation in each region each year. For each year, there is no region where all bonds are only rated by credit rating agencies with high (low) reputation. Moreover, there is no group of credit rating agencies with high (low) reputation only covering a certain region. Therefore, *Reputation_CRA* can only determine the differences between the high and low reputations of credit rating agencies instead of the differences between regions.

## Results and discussion

### Baseline regression analysis

The baseline regression results for **H1** and **H2** are listed in Table 8. *Risk Premium 5* and *Risk Premium 0* are set as the dependent variable in sequence. The year, region, and issuer fixed effects are included in the regression for robustness. Compared with the regression results without the interaction term (*RepFis*) shown in Columns (1) and (2) of Table 7, Columns (3) to (6) present those with *RepFis*. Meanwhile, Columns (3) and (4) only include issue-specific control variables, but columns (5) and (6) include issue-specific and issuer-specific control variables.

In all the preceding regressions, coefficients of *Reputation_CRA* are negative and statistically significant at the conventional levels, demonstrating that the credit rating agency reputation is negatively related to local government bonds' risk premiums. Hence, **H1** is supported. This result also proves that the information certification role of rating agencies can function independently in the local government bond market when noise from the information revelation role is removed.

In addition to statistical significance, the discussion of economic significance is equally important. In Column (1), coefficient of *Reputation_CRA* is -0.6136 and significant at the 1% level. As shown in the mean value of risk preium (28.6754 bp) in Table 5, the regression result implies that, on average, hiring a prestigious credit rating agency can reduce risk premium by 2.14%. More specifically, among the 7,941 research observations in our study, 2,982 local government bonds are rated by reputable credit rating agencies, and the total amount of their issue sizes is RMB13.39 trillion ($2.10 trillion). Overall, they are regarded to have saved

**Table 8. Baseline regression results.** This table reports the estimates of the baseline regression Eq (1).

| | Predicted sign | Dependent Variables | | | | | |
| --- | --- | --- | --- | --- | --- | --- | --- |
| | | Risk Premium 5 | Risk Premium 0 | Risk Premium 5 | Risk Premium 0 | Risk Premium 5 | Risk Premium 0 |
| | | (1) | (2) | (3) | (4) | (5) | (6) |
| Reputation_CRA | - | -0.6136*** (0.2370) | -0.5596** (0.2410) | -5.7837* (3.1190) | -10.9315*** (3.1751) | -6.8213** (3.2781) | -12.3012*** (3.3297) |
| Fiscal Transparency | - | -1.0418** (0.4479) | -0.8226* (0.4589) | -2.8153*** (0.6541) | -3.3828*** (0.6614) | -2.8147*** (0.6507) | -3.3497*** (0.6586) |
| RepFis | + | | | 1.5471* (0.8046) | 2.7923*** (0.8156) | 1.8592** (0.8509) | 3.1622*** (0.8609) |
| Maturity | + | 0.0632*** (0.0124) | 0.0576*** (0.0113) | 0.0466*** (0.0127) | 0.0417*** (0.0114) | 0.0593*** (0.0129) | 0.0547*** (0.0116) |
| Issue Size | + | 0.2342** (0.0927) | 0.2018** (0.0931) | 0.2247** (0.0912) | 0.2007** (0.0915) | 0.2020** (0.0903) | 0.1662* (0.0904) |
| Issue Frequency | - | 1.2904*** (0.2704) | 1.2186*** (0.2747) | 0.3975 (0.3579) | -0.0308 (0.3745) | 0.3015 (0.3567) | -0.2053 (0.3729) |
| Bond Type | + | 1.3302*** (0.2439) | 1.0913*** (0.2456) | 1.3054*** (0.2406) | 1.1100*** (0.2412) | 1.2690*** (0.2390) | 1.0641*** (0.2394) |
| Sale Methods | + | 25.3776*** (0.3425) | 24.7330*** (0.3630) | 25.2230*** (0.3451) | 24.5417*** (0.3663) | 25.3610*** (0.3388) | 24.6923*** (0.3601) |
| GDP Year | - | -0.5173** (0.2221) | -0.286 (0.2311) | | | 3.5619 (3.2057) | 0.7516 (3.2877) |
| GDP per capita | - | 1.4361** (0.5928) | 1.1869** (0.6008) | | | 0.8294 (0.9581) | 1.4727 (0.972) |
| GDP Growth Rate | - | -0.1141** (0.0497) | 0.0183 (0.0509) | | | 0.0534 (0.0596) | 0.1436** (0.0602) |
| FAI Growt hRate | + | 0.0532*** (0.0152) | 0.0550*** (0.0148) | | | 0.0811*** (0.0169) | 0.0799*** (0.0167) |
| Debt Ratio | + | 0.0115*** (0.0021) | 0.0118*** (0.0022) | | | 0.0119*** (0.0037) | 0.0140*** (0.0037) |
| Public Revenue per capita | - | -1.4508*** (0.4364) | -1.2327*** (0.4450) | | | 5.2940*** (0.9620) | 6.8687*** (0.9590) |
| Public Revenue Growth Rate | - | -0.0267 (0.0216) | -0.0648*** (0.0217) | | | -0.0933*** (0.0237) | -0.1212*** (0.0242) |
| Tbill | + | 4.2259*** (0.3375) | 4.2201*** (0.3355) | | | 4.2722*** (0.3399) | 4.2533*** (0.3343) |
| Constant | | 5.2462 (6.1783) | 4.2243 (6.3320) | 24.5160*** (2.6851) | 29.6191*** (2.7027) | -76.6889*** (28.3383) | -65.0953** (28.5797) |
| Year Dummies | | Included | Included | Included | Included | Included | Included |
| Region Dummies | | Included | Included | Included | Included | Included | Included |
| Issuer Dummies | | Excluded | Excluded | Included | Included | Included | Included |
| Adjusted R-squared | | 0.5840 | 0.5730 | 0.5960 | 0.5880 | 0.6060 | 0.5990 |
| No. of observations | | 7941 | 7941 | 7941 | 7941 | 7941 | 7941 |

A superscript *, ** or *** indicates significance at the 10%, 5% or 1% levels, respectively.

RMB821.61 million (RMB13.39 trillion*0.6136*0.0001), which is more than the issue sizes of 1911 bonds observed in our study. Therefore, our study confirms that hiring reputable credit rating agencies has a considerable saving effect on borrowing costs. That is, the debt burden of local governments is relieved correspondingly. Hence, it is conducive to the long-term sustainability of issuers' public finance.

This finding resonates with Livingston, Poon, and Zhou [7] and Hu, Shi, Wang, and Yu [11], who reported 18 bp and 4.8 bp yield reductions, respectively, for bond issues rated by

prestigious credit rating agencies based on the China's corporate bonds sample. This empirical result suggests that China's local government bond investors do differentiate among credit rating agencies based on their perceived reputation. Ratings issued by more reputable credit rating agencies provide a stronger certification of the bond's credit quality. Nevertheless, the degree of risk premium reduction is less than that in the corporate bond market, implying that the local government bond market has more political issues and/or higher information asymmetry. When the dependent variable is replaced with *Risk Premium 0*, the result in Column (2) shows high similarity to that in Column(1). Therefore, the baseline regression can be certified to be robust preliminarily.

The results in Columns (3) to (6) present the moderating effect of fiscal transparency on the impact of credit rating agency reputation on risk premiums. Coefficients of interaction term (*RepFis*) are positive and statistically significant. This result indicates that issuers' voluntary disclosure of their financial information weakens the negative relationship between credit rating agency reputation and risk premiums. When comparing local governments perceived at low-level fiscal transparency by market participants with those perceived at high-level fiscal transparency, when both types of issuers hire any prestigious credit rating agency, the former can issue bonds at lower risk premiums, thereby obtaining a better reduction effect than the latter. Hence, *H2* is supported. Moreover, credit rating agency reputation helps local governments with low fiscal transparency reduce more risk premiums.

To offer a markedly intuitionistic illustration of the marginal effect of *Reputation_CRA* on local government bonds' risk premiums varying with the issuers' fiscal transparency (*Fiscal Transparency*), the moderating effect based on the regression result in Column (5) is plotted in Fig 1.

When issuers' fiscal transparency is low (in the left range of the intersection of the blue and red lines), risk premiums of local government bonds by hiring a reputable credit rating agency (red) are significantly lower than the risk premiums rated by a low-reputation credit rating agency (blue). However, when issuers' fiscal transparency is high (in the right range of the intersection of the blue and red lines), the bonds' risk premiums rated by a high-reputation

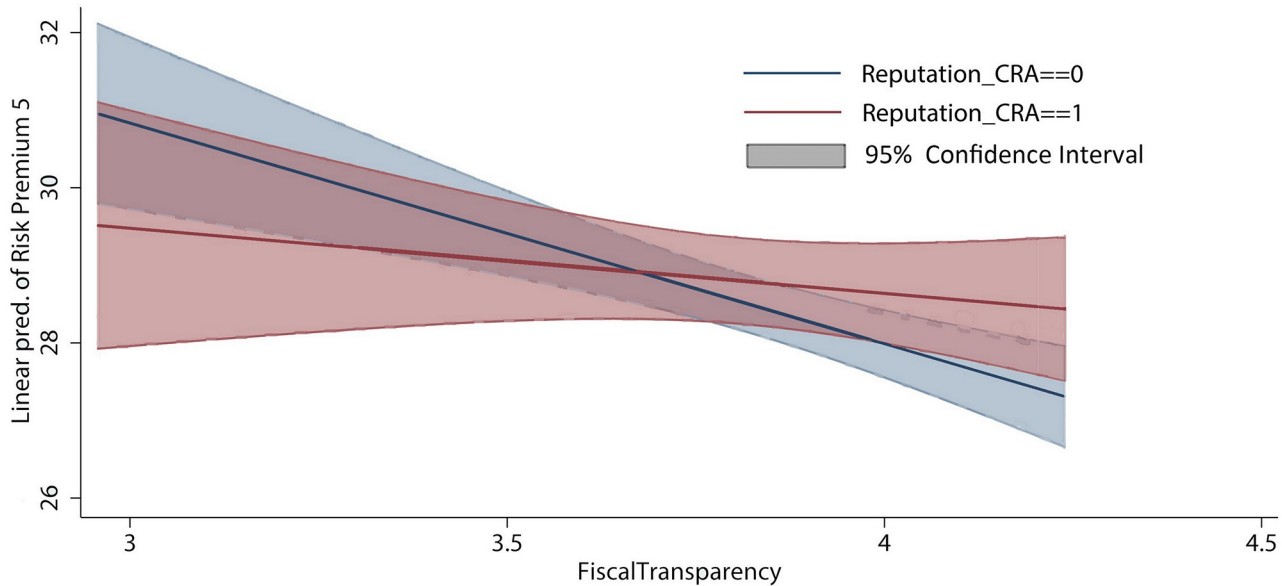

**Fig 1. Moderating effect of fiscal transparency.**

credit rating agency are higher than those rated by a low-reputation credit rating agency. One possible reason for this result is that local governments with high fiscal transparency have already obtained relatively low borrowing costs. Hence, they have limitation in achieving further reduction of risk premiums owing to employing prestigious information intermediaries. By contrast, local governments with low fiscal transparency should immediately employ reputable credit rating agencies to relieve their debt burden. Moreover, information transparency of the entire market will be effectively improved, which will contribute to the sustainable development of the local government bond market.

Table 8 also shows the predicted sign of each variable. Consistent with our expectation, coefficients of *Maturity* are significantly positive, indicating that the longer maturity, the higher the risk premium. Coefficients of *Issue Size* are significantly positive and consistent with the findings of Hu, Shi, Wang, and Yu [11]. Significantly positive coefficients of *Issue Frequency* are unexpected. Peng and Brucato [50] argued that when one issuer frequently enters the market, its degree of information disclosure should accumulate gradually, resulting in low risk premium. However, positive coefficients of *Issue Frequency* are likely caused by the higher frequent issuance accompanying a higher debt burden. For different types of bonds (*Bond Type*), coefficients are significantly positive in all columns, suggesting higher risk premiums on revenue bonds than general obligation bonds. This result is consistent with Butler, Fauver, and Mortal [17]. Coefficients of *Sale Methods* are also significantly positive, indicating that private placement costs more than the public offering sale method. On average, local governments have to pay an additional yield of over 25 bp when their bonds are sold by private placement.

For issuer-specific variables, coefficients of *GDP Year* and *GDP Growth Rate* are negative and significant at the 5% level, as shown in Column (1). This result suggests that high-level economic development is beneficial in reducing local government bonds' risk premiums. However, positive coefficient of *GDP per capita* contracts with our expectation. Coefficients of *FAI Growth Rate* and *Debt Ratio*, which are significant at the 1% level as shown in Columns (1) and (2), suggest that issuers' debt level is positively related to local government bonds' risk premiums. Similarly, negative coefficients of *Public Revenue per capita* and *Public Revenue Growth Rate* (significant at the 1% level in Columns (1) and (2)) imply that the stronger the revenue capacity of local governments, the lower the local government bond's risk premium. Consistent with our expectation, *Tbill* is reported to be positively related to the risk premiums of local government bonds at issuance in Columns (1), (2), (5) and (6).

## Robustness checks

**Endogeneity concern: Heckman two-stage model.** The findings in the baseline regressions support the idea that local government bonds rated by more reputable credit rating agencies have lower risk premiums. A concern with this finding is potential selection bias. Local governments' choice of credit rating agencies may not be random but possibly driven by unobservable issuer characteristics. If these issuer characteristics are also correlated with bond yields, then the results in Table 8 may be biased. Therefore, we employ the Heckman two-stage sample selection model [54] to address the potential selection bias, referring to some previous studies showing that the method can address the endogeneity issue [55, 56].

The first stage contains a probit model of the local government's choice of two types of credit rating agencies: credit rating agencies with high and low reputations. Dependent variable in this stage is set to 1 for bonds rated by three more reputable credit rating agencies and 0 otherwise. One important explanatory variable in the first stage probit model is *Reputation_-CRA_MarketShare*, provincial market shares of the more reputable credit rating agencies

(CCXI_Moody, Lianhe_Fitch, or Brilliance_S&P) in the corporate bond market of the previous year. The variable is defined as follows: for each issuer of local government bonds, the percentage of corporate bonds rated by more reputable credit rating agencies to the total corporate bonds issued by the companies are registered in the corresponding province in the previous year. Evidently, the corporate bond and local government bond markets are different from each other. Therefore, *Reputation_CRA_MarketShare* in the corporate bond market is less likely to be directly correlated with error terms of the local government bonds' risk premium regression [7]. Thus, that *Reputation_CRA_MarketShare* is excluded in the second stage regression of the local government bond risk premium meets the exclusion restriction criteria. The results of regressions based on the Heckman two-stage model are listed in Table 9.

Column (1) of Table 9 reports the probit model results of choosing credit rating agencies. Coefficient of *Reputation_CRA_MarketShare* is positive and significant at the 1% level. That is, if credit rating agencies with high reputaion have large market shares in the corporate bond market in one province, then these agencies are likely to be chosen by the corresponding local government to rate its local government bonds.

In the second stage, *Inverse Mills Ratio* is calculated based on the results from the first stage probit selection model. Thereafter, the model of local government bond risk premium, Eq (1), is regressed with *Inverse Mills Ratio* as an additional explanatory variable. Column (2) of Table 9 reports the results of the second-stage regression. Coefficient of *Inverse Mills Ratio* is positive but not statistically significant, implying that potential selection bias is not a severe problem in this study. Moreover, coefficient of *Reputation_CRA* remains negative and significant at the 1% level, and the coefficient of interaction term (*RepFis*) remains positive and significant at the 1% level as those in the baseline regression. Therefore, the results are robust with consideration of potential selection bias, and provide further evidence to support **H1** and **H2**.

**Endogeneity concern: Difference-in-differences regression.**   A possibility of bias exists, in which the reported reduction effects on risk premiums from prestigious credit rating agencies could be driven by the fact that local governments with high creditworthiness and low risk premiums in their bonds tend to hire reputable credit rating agencies. We refer to He, Li, and Luo [26] and He, Ren, and Taffler [57] and take advantage of the opening up of the credit rating industry to foreign agencies as exogenous shock and conduct a DID OLS regression, stated as the Eq (2), to solve the potential endogenous problem.

$$Risk\ Premium_{i,p,t} = \gamma_0 + \gamma_1 Reputation\_CRA_{i,p,t} + \gamma_2 Reputation\_CRA_{i,p,t} * Post_t +$$
$$\gamma_3 * Post_t + Control\ Variables + Year\ Dummies + Issuer\ Dummies + \delta_{i,p,t} \tag{2}$$

The dependent variable is *Risk Premium*, which is the same as in the baseline regression. The treatment indicator variable is the core explanatory variable (*Reputation_CRA*) in the baseline regression, equal to 1 for credit rating agencies with international partners and 0 for local agencies. Accompanied by three overseas credit rating agencies with high reputation(i.e., S&P, Fitch, and Moody) directly enter the China's bond market, they are expected to take more extensive penetration and guidance for their partners (Brilliance_S&P, Lianhe_Fitch, and CCXI_Moody) in China. Hence, dividing credit rating agencies into treatment and control groups according to their ownership is reasonable.

The time indicator variable, *Post*, equals 1 if a local government bond is issued in the year in the post-opening period (i.e., 2019–2021), and 0 if the local government bond is issued in the pre-opening period (i.e., 2015–2018). Although China opened up its credit rating industry to foreign credit rating agencies in March 2018, S&P became the first foreign credit rating agency to set up a wholly owned subsidiary in January 2019. This development proves that the

**Table 9. Regression of risk premiums with Heckman correction.** This table reports the estimates of Heckman two-stage model to control for potential sample selection bias. The first stage reports the results of probit model, while the second stage shows the results of the baseline regression model.

| Dependent variable: Risk Premium 5 | First stage | Second stage |
|---|---|---|
| | Probit model of choosing CRA | Risk premiums of local government bonds |
| Reputation_CRA | | -10.0667*** |
| | | (2.5152) |
| Reputation_CRA_MarketShare | 0.0064*** | |
| | (0.0018) | |
| Inverse Mills ratio | | 0.4405 |
| | | (1.1240) |
| Fiscal Transparency | -1.0017*** | -2.1745** |
| | (0.0624) | (0.9137) |
| RepFis | | 2.5092*** |
| | | (0.6538) |
| Maturity | -0.0014 | 0.0603*** |
| | (0.0027) | (0.0124) |
| Issue Size | 0.0196 | 0.2456*** |
| | (0.0139) | (0.0937) |
| Issue Frequency | 0.3048*** | 1.3008*** |
| | (0.0450) | (0.3487) |
| Bond Type | -0.0034 | 1.3445*** |
| | (0.0385) | (0.2435) |
| Sale Methods | 0.0260 | 25.4416*** |
| | (0.0556) | (0.3417) |
| GDP Year | 0.4622*** | -0.4016 |
| | (0.0447) | (0.4406) |
| GDP per capita | -2.4696*** | 0.8126 |
| | (0.1620) | (1.7183) |
| GDP Growth Rate | -0.0681*** | -0.1011 |
| | (0.0072) | (0.0795) |
| FAI Growth Rate | 0.0077*** | 0.0575*** |
| | (0.0028) | (0.0159) |
| Debt Ratio | 0.0061*** | 0.0136** |
| | (0.0004) | (0.0057) |
| Public Revenue per capita | 0.9081*** | -1.2584* |
| | (0.1057) | (0.7381) |
| Public Revenue Growth Rate | -0.0263*** | -0.0267 |
| | (0.0030) | (0.0262) |
| Tbill | -0.1271** | 4.2454*** |
| | (0.0573) | (0.3505) |
| Constant | 15.4460*** | 12.1838 |
| | (0.9832) | (10.5492) |
| Year Dummies | Included | Included |
| Region Dummies | Included | Included |
| Pseudo R-squared | 0.2139 | |
| Adjusted R-squared | | 0.5850 |
| Observations | 7941 | 7941 |

A superscript *, ** or *** indicates significance at the 10%, 5% or 1% levels, respectively.

real effect of this policy on the existing agencies takes time to materialize. Therefore, we define our post-event period starting from 2019.

In this regression, we focus on coefficients of *Reputation_CRA* and *Reputation_CRA*Post*, and the latter captures the effect of the opening-up policy on the reputation certification of

**Table 10. Univariate tests of covariate balance for propensity-score matching.** This table reports descriptive statistics of the covariates for the sample of treatment bonds and the sample of control bonds. The results of the two-sample tests of mean and of the standardized bias for the covariates are provided for both unmatched and matched samples.

| Variables | Matching status | No. of bonds | Mean for treatment bonds | Mean for control bonds | Standardized bias (%) | t-stat. |
|---|---|---|---|---|---|---|
| Maturity | Unmatched | 7941 | 9.4658 | 9.3880 | 1.2 | 0.51 |
| | Matched | 4100 | 9.4738 | 9.3320 | 2.1 | 0.84 |
| Issue Size | Unmatched | 7941 | 7.7533 | 7.6090 | 10.8 | 4.64*** |
| | Matched | 4100 | 7.7524 | 7.7211 | 2.3 | 0.94 |
| Issue Frequency | Unmatched | 7941 | 3.8284 | 3.7517 | 13.9 | 5.96*** |
| | Matched | 4100 | 3.8262 | 3.8327 | -1.2 | -0.45 |
| Bond Type | Unmatched | 7941 | 0.6519 | 0.6278 | 5.0 | 2.17** |
| | Matched | 4100 | 0.6516 | 0.6340 | -2.6 | -1.01 |
| Sale Methods | Unmatched | 7941 | 0.1365 | 0.1406 | -1.2 | -0.51 |
| | Matched | 4100 | 0.1368 | 0.1371 | -0.1 | -0.04 |
| Debt Ratio | Unmatched | 7941 | 169.90 | 167.66 | 2.5 | 1.09 |
| | Matched | 4100 | 169.93 | 168.67 | 1.4 | 0.54 |

A superscript *, ** or *** indicates significance at the 10%, 5% or 1% levels, respectively.

credit rating agencies. The two coefficients provide evidence for the robustness of baseline regression and the ruleout of potential endogeneity of the core explanatory variable, respectively.

To remediate the potential bias, that is, the systemic differences in bond characteristics between bonds rated by high-reputation credit rating agencies and those rated by low-reputation credit rating agencies, the PSM method is used to match the bonds in the treatment group to the control group based on the nearest neighbor propensity score estimated from a set of selected bond characteristics. After this preprocessing, we obtain the final sample for DID OLS regression, consisting of 4100 observations. By referring to previous literature [26, 57, 58], the validity of PSM is tested by the univariate test of covariate balance. The results are listed in Table 10. It can be found that, in all matched groups, there is no statistically significant difference in the mean values of covariates between the treatment and control groups (as shown in the statistical characteristic in t-stat)., and the standardized bias are under 10%. The preceding features demonstrate that the post-matching samples meet the requirement of covariate balance, and PSM can significantly reduces the differences between bonds in the treatment and control groups.

Table 11 shows the DID OLS regression results based on the 4100 observations obtained from PSM. Coefficients ($\gamma_2$) of interaction terms (*Reputation_CRA*Post*) are negative and statistically significant at the conventional levels. The interaction term coefficient (*Reputation_CRA*Post*) is –2.1285 (-1.5364), suggesting that the opening-up policy results in a substantial reduction in *Risk Premium 5* (*Risk Premium 0*), which is about 14.95% (10.84%) of one standard deviation of *Risk Premium 5* (*Risk Premium 0*) (see Table 5). Thus, the interaction term is statistically and economically meaningful.

On the one hand, this statistical and economic significance proves that the opening-up policy, which is an exogenous shock, has a significant impact on the rating industry in China, thereby effectively alleviating the potential endogenous problems in this study. On the other hand, comparing the risk premiums of local government bonds rated by the credit rating agencies identified with high reputation (*Reputation_CRA*=1) and those with low reputation (*Reputation_CRA*=0), the negative values of $\gamma_2$ means that, after the foreign partners of credit rating

**Table 11. Difference-in-differences OLS regression results.** This table reports the regression results based on Eq (2).

|  | Dependent Variable: Risk Premium 5 | Dependent Variable: Risk Premium 0 |
|---|---|---|
| Reputation_CRA | -15.2346*** | -14.7596*** |
|  | (5.0955) | (5.1544) |
| Reputation_CRA*Post | -2.1285*** | -1.5364* |
|  | (0.8047) | (0.8144) |
| Post | 3.1792 | 2.8312 |
|  | (2.2212) | (2.2096) |
| Fiscal Transparency | -4.1884*** | -4.2107*** |
|  | (1.0694) | (1.0842) |
| RepFis | 4.5849*** | 4.3256*** |
|  | (1.4009) | (1.4159) |
| Maturity | 0.0581*** | 0.0528*** |
|  | (0.0177) | (0.0160) |
| Issue Size | 0.3614*** | 0.2888** |
|  | (0.1213) | (0.1228) |
| Issue Frequency | 0.5527 | 0.0279 |
|  | (0.5134) | (0.5396) |
| Bond Type | 1.1440*** | 1.0378*** |
|  | (0.3299) | (0.3318) |
| Sale Methods | 25.2088*** | 24.9159*** |
|  | (0.4701) | (0.4990) |
| GDP Year | 4.5031 | 0.9336 |
|  | (4.7728) | (4.8052) |
| GDP per capita | 1.0783 | 1.7562 |
|  | (1.5432) | (1.5193) |
| GDP Growth Rate | 0.0485 | 0.1176 |
|  | (0.0815) | (0.0833) |
| FAI Growth Rate | 0.0812*** | 0.0850*** |
|  | (0.0250) | (0.0249) |
| Debt Ratio | 0.0110** | 0.0140** |
|  | (0.0055) | (0.0054) |
| Public Revenue Growth Rate | 5.7585*** | 8.4974*** |
|  | (1.3447) | (1.3586) |
| Public Revenue per capita | -0.0675** | -0.1039*** |
|  | (0.0331) | (0.0336) |
| Tbill | 3.8502*** | 3.9256*** |
|  | (0.4526) | (0.4473) |
| Constant | -86.7606** | -78.2054* |
|  | (42.1428) | (42.3619) |
| Year Dummies | Included | Included |
| Issuer Dummies | Included | Included |
| Adjusted R-squared | 0.6120 | 0.6060 |
| Observations | 4100 | 4100 |

A superscript *, ** or *** indicates significance at the 10%, 5% or 1% levels, respectively.

agencies with high reputation entered into China's local government bond market, the former is further lower than the latter. The reason for the further reduction of the risk premium is that these foreign partners are currently able to guide the rating work more extensively and directly than before the opening-up policy, further enhancing the recognition of their good reputation by market participants.

Coefficient of *Reputation_CRA* is negative and statistically significant, further supporting *H1*. Similarly, coefficient of *RepFisit* is positve and statistically significant, supporting *H2*. The

preceding features of coefficients are consistent with the baseline regression results. Hence, the results can also prove the robustness of the aforementioned baseline regression.

Moreover, there is an important assumption (parallel trends assumption) behind the DID regression, which requires that the outcome variable (*Risk Premium*) should have similar trends between the treatment and control groups in the absence of the treatment event. To test this assumption, we refer to the literature [59], and add three interaction terms (i.e. *Reputation_CRA*Year2016*, *Reputation_CRA*Year2017*, *Reputation_CRA*Year2018*) in the DID OLS regression model. *Year2016* to *Year2018* are the year dummies for 2016 to 2018, respectively. The results are presented in Table 12.

All coefficients of the three interaction terms (*Reputation_CRA*Year2016*, *Reputation_CRA*Year2017*, and *Reputation_CRA*Year2018*) are statistically insignificant, and these results indicate that there are no differences between the treatment and control groups before the opening event. Thus, the parallel trends assumption holds for the DID OLS regression, supporting that the preceding results and analysis of DID OLS regression are reliable.

**Endogeneity concern: Machine learning method.** According to the framework of counterfactual inference provided by Rubin (1974) [60], the main work of causal inference is to construct a counterfactual set of data. Owing to the unobservability of counterfactors, which is a fundamental problem of causal inference [61], the limitation of traditional econometric methods, which only be able to extract limited linear or nonlinear features of existing data, makes themselves helpless in constructing a perfect set of counterfactual data. By contrast, the machine learning method is popular for its capacity to learn any complicated interrelation and to predict based on the training. Hence, it has the potential to be engaged in solving this problem. This research also encounters the same unobservable problem. In the baseline regression, there are 2982 bonds the issuers of which hire credit rating agencies with high reputation (*Reputation_CRA*=1). For these bonds, if their current risk premiums are regarded as the fact, then we can never obtain the counterfactual risk premiums in reality. That is, risk premiums when their issuers hire credit rating agencies with low reputation but all other conditions (issue- and issuer-specific features) remain unchanged. Therefore, we attempt to use machine learning to predict some counterfactual risk premiums.

With all variables of the 7941 bonds included in the baseline regression, we trained an optimizable tree model and optimizable ensemble of trees model, which are common in related fields, with the method of fivefold cross-validation in Regression Learner Application from MATLAB (MathWorks). For the optimizable tree model, the application provides a hyperparameter option that can be optimized, that is, minimum leaf size, from 1 to 3970 in this case. For the optimizable ensemble of trees model, including the minimum leaf size, the application also provides other hyperparameter options that can be optimized, including the ensemble method (Bag or LSBoost), number of learners (from 10 to 500), and number of predictors to sample (from 1 to 58 in this case). During optimization, if the ensemble method is LSBoost, then learning rate can also be optimized from 0.001 to 1. The results of the training are shown in Figs 2 and 3. For each observation, the closer the point to the line of perfect prediction, the better the training performs.

We use the aforementioned two trained models and select observations the issuers of which hire credit rating agencies with high reputation (*Reputation_CRA* = 1) and residuals between predicted and true responses are in the range of [-0.5, 0.5] to ensure the best prediction performance. We replace all values of *Reputation_CRA* in the selected set with 0 and all other variables remain unchanged to construct a new set of predictors (independent variables). This new set of predictors is used as input to the trained models to predict their counterfactual risk premiums. The statistics and predicted results of the selected samples are analyzed in Table 13.

**Table 12. Test of parallel trends assumption.** This table reports the results for the test of parallel trends assumption for difference-in-differences OLS regression.

| | Dependent Variable: Risk Premium 5 | Dependent Variable: Risk Premium 0 |
|---|---|---|
| Reputation_CRA | -16.1694*** | -15.1900*** |
| | (5.1494) | (5.2030) |
| Reputation_CRA*Year2016 | 1.2793 | 1.6460 |
| | (1.2809) | (1.3505) |
| Reputation_CRA*Year2017 | -2.1976 | -2.2799 |
| | (1.6101) | (1.7040) |
| Reputation_CRA*Year2018 | 0.9067 | 1.6594 |
| | (1.7523) | (1.7909) |
| Reputation_CRA*Post | -2.2094* | -1.2829 |
| | (1.2985) | (1.3645) |
| Post | 2.9922 | 2.4846 |
| | (2.3088) | (2.2952) |
| Fiscal Transparency | -4.0707*** | -3.9390*** |
| | (1.1020) | (1.1256) |
| RepFis | 4.8597*** | 4.3948*** |
| | (1.4507) | (1.4716) |
| Maturity | 0.0577*** | 0.0524*** |
| | (0.0177) | (0.0160) |
| Issue Size | 0.3426*** | 0.2674** |
| | (0.1207) | (0.1220) |
| Issue Frequency | 0.4701 | -0.0661 |
| | (0.5116) | (0.5375) |
| Bond Type | 1.1463*** | 1.0375*** |
| | (0.3289) | (0.3307) |
| Sale Methods | 25.1244*** | 24.8223*** |
| | (0.4758) | (0.5034) |
| GDP Year | 5.3075 | 1.8487 |
| | (4.8177) | (4.8514) |
| GDP per capita | 0.8332 | 1.4552 |
| | (1.5729) | (1.5446) |
| GDP Growth Rate | 0.0711 | 0.1462* |
| | (0.0813) | (0.0835) |
| FAI Growth Rate | 0.0885*** | 0.0931*** |
| | (0.0251) | (0.0250) |
| Debt Ratio | 0.0102* | 0.0133** |
| | (0.0055) | (0.0055) |
| Public Revenue per capita | 5.8182*** | 8.5088*** |
| | (1.3318) | (1.3411) |
| Public Revenue Growth Rate | -0.0734** | -0.1101*** |
| | (0.0334) | (0.0338) |
| Tbill | 3.8698*** | 3.9379*** |
| | (0.4506) | (0.4453) |
| Constant | -92.9657** | -84.9721** |
| | (42.3139) | (42.5802) |
| Year Dummies | Included | Included |
| Issuer Dummies | Included | Included |
| Adjusted R-squared | 0.6130 | 0.6070 |
| Observations | 4100 | 4100 |

A superscript *, ** or *** indicates significance at the 10%, 5% or 1% levels, respectively.

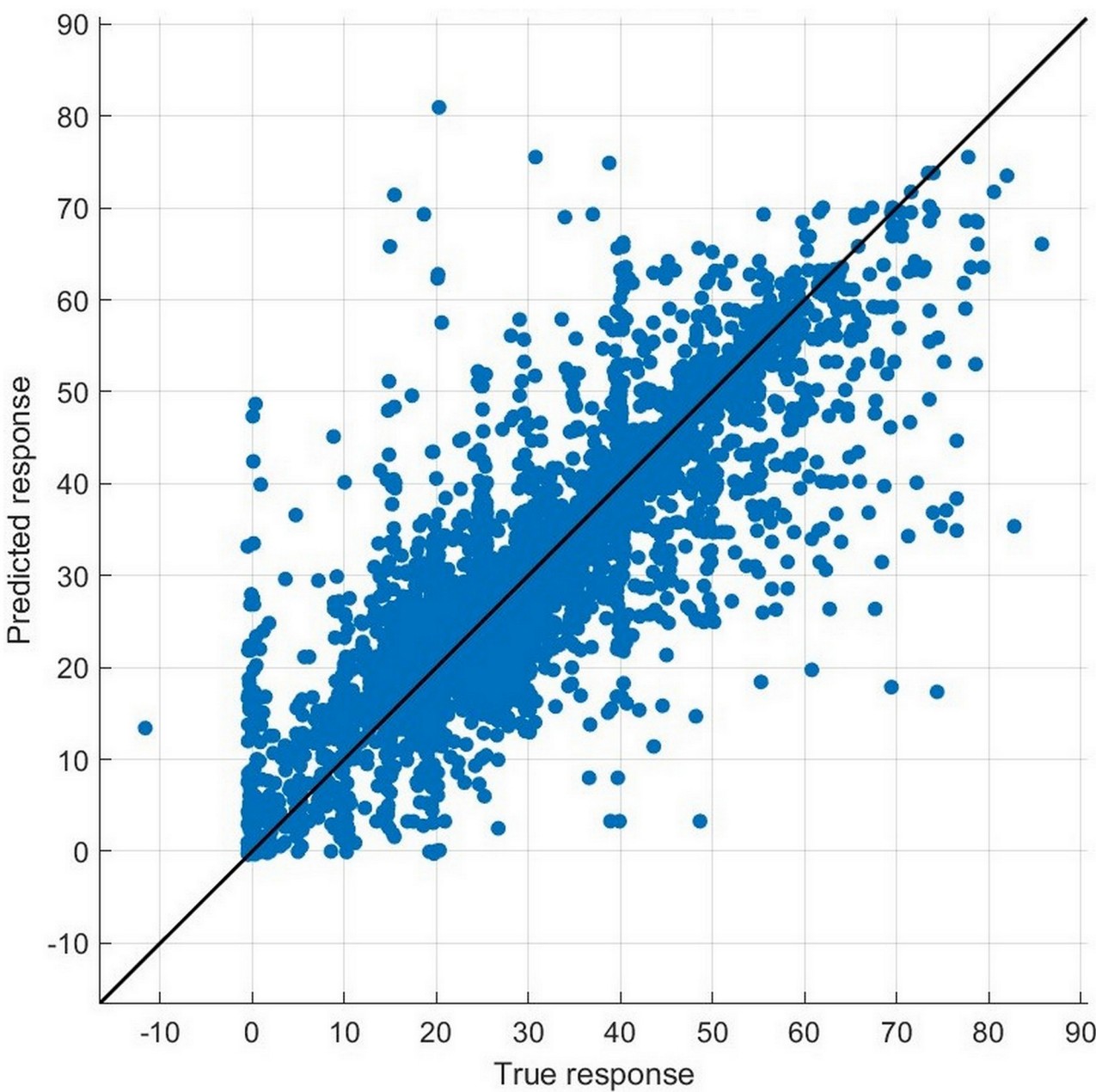

**Fig 2. Predicted response versus true response of the trained optimizable tree model.**

To analyze the results, referring to the evaluation criteria of PSM, as shown in Table 13, the t-test between risk premiums of the factual and counterfactual sets is completed. Moreover, the result shows a significant difference between the two sets of risk premiums. Note that we only replaced the value of the core explanatory variable with 0 during the construction of counterfactual set and the dependent variable changed significantly as expected. Therefore, the causal relationship between *Reputation_CRA* (the cause) and risk premium (the effect) can be deemed tenable. That is, this result can alleviate the potential endogenous problems in this study as well. In addition, we calculate the difference in the average risk premiums between

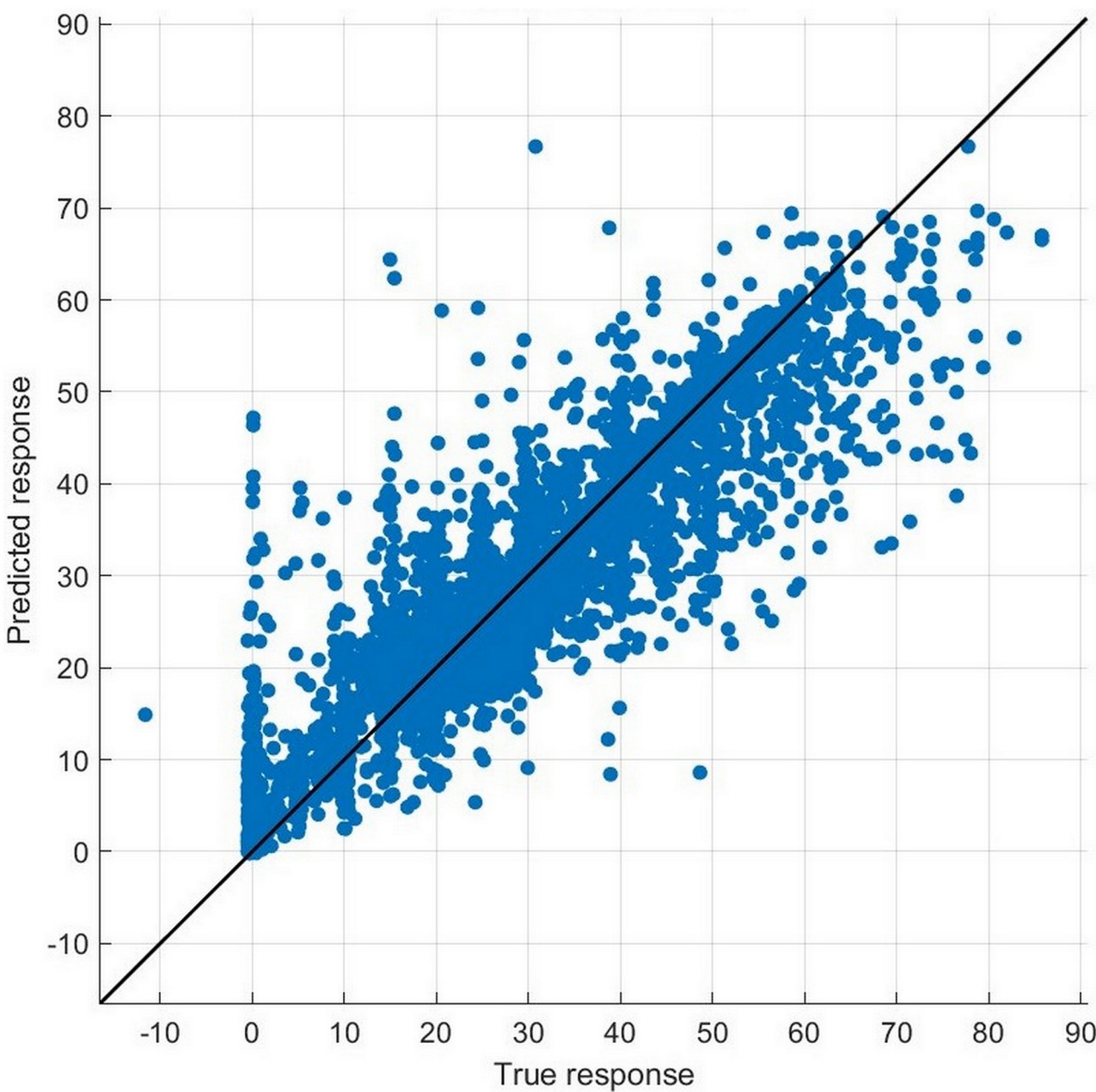

**Fig 3. Predicted response versus true response of the trained optimizable ensemble of tree model.**

the factual and counterfactual sets. Note that both values are negative. By considering the significant P-value of the t-test together, hiring credit rating agencies with high reputation can reduce risk premium. Thus, **H1** is supported.

**Robustness: Excluding observations with negative risk premium.** The entire sample includes observations with negative risk premium, implying the existence of political intervention from issuers. These observations can be sample outliers that may cause estimation bias. Therefore, this section repeats the baseline analysis by excluding 186 observations with negative *Risk Premium 5* and 206 observations with negative *Risk Premium 0*. The results are

**Table 13. The validation of endogeneity based on the machine learning model.** This table reports the optimized hyperparameters of the two machine learning models (optimizable tree model and optimizable ensemble of trees model) and the statistics of counterfactual sets constructed by these two models. The last row reports the p-value of the t-test on the differences in risk premiums in the factual and counterfactual sets.

| Model of machine learning | Optimizable tree | Optimizable ensemble of trees |
|---|---|---|
| Root mean square error | 6.3399 | 5.5068 |
| R-squared | 0.80 | 0.85 |
| **Optimized hyperparameters** | | |
| Ensemble method | N/A | Bag |
| Minimum leaf size | 1 | 8 |
| Number of learners | N/A | 30 |
| Number of predictors to sample | N/A | 1 |
| Optimizer | Bayesian optimization | Bayesian optimization |
| **Statistics** | | |
| Number of obervations in the counterfactual set | 1365 | 1281 |
| Proportion of the observations of the counterfactual set to the total | 45.77% | 42.96% |
| Mean of risk premium in the factual set | 28.2757 bp | 28.1032 bp |
| Mean of risk premium in the counterfactual set | 28.3760 bp | 28.1593 bp |
| Difference of mean of risk premium between factualand counterfactual set | -0.103 | -0.0561 |
| P-value of t-test | 0.0565 | 0.0139 |

presented in Table 14. Including the demonstration of the robustness, the main features of the results remain consistent with all the preceding results. Therefore, *H1* and *H2* are supported.

**Robustness: Subperiod analysis.**   Following the COVID-19 pandemic, many governments globally have adopted looser fiscal and monetary policies. The scale of local government bonds in China also has a substantial growth after 2019. Accordingly, relevant policies may influence the risk premiums of local government bonds issued in 2020–2021. To alleviate the impact of COVID-19 on our research conclusions, we repeat the baseline regression by excluding data from 2020–2021. The regression results based on 4846 observations from 2015 to 2019 are presented in Table 15. The main features of the results of this subperiod analysis remain consistent with the preceding full-period results. Therefore, this analysis proves the robustness and support *H1* and *H2*.

## Conclusion

Effective management of local government debt is important in assuring the sustainability of public finance and economic growth. Given that all local government bonds in China obtain the same ratings, existing studies have directly disregarded the impact of credit rating agencies on pricing [39, 51]. In addition, most studies have investigated the sustainability of local government debt from the perspectives of debt scale controlling and early risk warning. However, this paper regards credit rating agency as an important information intermediary in the bond market. In-depth research on their role and influence on local government bonds' pricing in China should be conducted. In this context, this paper examines the relationship between credit rating agency's reputation and local government bond risk premiums. Accordingly, risk premiums of bonds rated by more reputable credit rating agencies are significantly lower. Meanwhile, issuers with lower fiscal transparency level obtain more risk premium reductions

**Table 14. Regression results by excluding negative risk premium observations.** This table reports the estimates of the baseline regression Eq (1), excluding negative risk premium observations, with robust standard error.

| | Dependent Variable: Risk Premium 5 | Dependent Variable: Risk Premium 0 |
|---|---|---|
| Reputation_CRA | -5.8774* | -6.4877** |
| | (3.2222) | (3.1507) |
| Fiscal Transparency | -2.1889*** | -1.9326*** |
| | (0.6452) | (0.6244) |
| RepFis | 1.5622* | 1.6458** |
| | (0.8384) | (0.8109) |
| Maturity | 0.0525*** | 0.0489*** |
| | (0.0125) | (0.0110) |
| Issue Size | 0.1755** | 0.0566 |
| | (0.0880) | (0.0859) |
| Issue Frequency | -0.0463 | -0.7141** |
| | (0.3484) | (0.3522) |
| Bond Type | 1.1344*** | 0.7120*** |
| | (0.2329) | (0.2261) |
| Sale Methods | 23.9969*** | 22.8660*** |
| | (0.3262) | (0.3380) |
| GDP Year | 3.3839 | -0.3653 |
| | (3.2437) | (3.1923) |
| GDP per capita | 0.3095 | 0.7237 |
| | (0.9887) | (0.9868) |
| GDP Growth Rate | 0.0140 | 0.0734 |
| | (0.0589) | (0.0572) |
| FAI Growth Rate | 0.0860*** | 0.0829*** |
| | (0.0165) | (0.0162) |
| Debt Ratio | 0.0095*** | 0.0082** |
| | (0.0037) | (0.0035) |
| Public Revenue per capita | 4.9147*** | 5.7803*** |
| | (0.9443) | (0.9196) |
| Public Revenue Growth Rate | -0.0908*** | -0.1149*** |
| | (0.0235) | (0.0237) |
| Tbill | 3.4321*** | 3.4444*** |
| | (0.3298) | (0.3067) |
| Constant | -61.9874** | -33.3729 |
| | (28.1619) | (27.0550) |
| Year Dummies | Included | Included |
| Region Dummies | Included | Included |
| Issuer Dummies | Included | Included |
| Adjusted R-squared | 0.6010 | 0.6090 |
| No. of observations | 7755 | 7735 |

A superscript *, ** or *** indicates significance at the 10%, 5% or 1% levels, respectively.

on their bonds by hiring prestigious rating agencies. This paper differs from the previous literature in two aspects. On the one hand, we draw attention to the reputation certification effect of the credit rating agency in the local government bond market. This market has high information asymmetry and complex political issues owing to the issuers' unique nature [17]. Therefore, evidence from other bond markets cannot be directly applied to this market. On the other hand, it is able to disentangle two information roles of the credit rating agency. Although some latest studies [7, 11, 45] have provided evidence on the corporate bond

**Table 15. Regression results based on the subperiod sample.** This table reports the estimates of the baseline regression Eq (1) from 2015 to 2019, with robust standard error.

| | Dependent Variable: Risk Premium 5 | Dependent Variable: Risk Premium 0 |
|---|---|---|
| Reputation_CRA | -9.3151*** | -11.7342*** |
| | (3.5313) | (4.5252) |
| Fiscal Transparency | -2.8774*** | -4.2322*** |
| | (0.6261) | (0.8571) |
| RepFis | 2.4667** | 2.8455** |
| | (0.9646) | (1.2326) |
| Maturity | 0.2314*** | 0.2394*** |
| | (0.0325) | (0.0350) |
| Issue Size | 0.3833** | 0.2649* |
| | (0.1495) | (0.1507) |
| Issue Frequency | 3.8958*** | 1.6502*** |
| | (0.4517) | (0.6159) |
| Bond Type | 1.9734*** | 1.5948*** |
| | (0.3296) | (0.3339) |
| Sale Methods | 25.5793*** | 24.7677*** |
| | (0.3430) | (0.3682) |
| GDP Year | -0.9763*** | -1.8641 |
| | (0.2969) | (6.0132) |
| GDP per capita | 2.6518*** | 0.9399 |
| | (0.8181) | (1.1708) |
| GDP Growth Rate | -0.0280 | 0.1171 |
| | (0.0722) | (0.1002) |
| FAI Growth Rate | 0.0112 | 0.0928*** |
| | (0.0226) | (0.0284) |
| Debt Ratio | 0.0078** | 0.0360*** |
| | (0.0031) | (0.0102) |
| Public Revenue per capita | -2.9156*** | 7.2940*** |
| | (0.6496) | (1.4388) |
| Public Revenue Growth Rate | -0.1113*** | -0.1075*** |
| | (0.0306) | (0.0358) |
| Tbill | 10.7009*** | 11.3094*** |
| | (0.8910) | (0.9540) |
| Constant | -8.1606 | -64.0723 |
| | (8.3752) | (56.8952) |
| Year Dummies | Included | Included |
| Region Dummies | Included | Included |
| Issuer Dummies | Excluded | Included |
| Adjusted R-squared | 0.6020 | 0.6040 |
| No. of observations | 4846 | 4846 |

A superscript *, ** or *** indicates significance at the 10%, 5% or 1% levels, respectively.

markets, they cannot fully exclude the influence of information revelation role to test the effect of rating agency's reputation certification.

By providing new empirical evidence to support the reputation certification hypothesis established by Booth and Smith [12] in the context of local government bond markets, findings of this paper also have several implications for participants in this market. First, existing domestic and potential foreign investors can rely on the credit rating agency's reputation to complement credit risk analysis. The reason is that these reputable credit rating agencies have more stringent rating standards and provide more reliable information. Second, issuers (local

governments) can lower borrowing costs by changing credit rating agencies with low reputation to those with high reputation, specifically for those issuers perceived as low transparency. According to our empirical results, the economical significance (the reduction effect on borrowing costs) for issuers is considerable. Third, regulators should highlight the supervision of credit rating agencies because of their substantial impact on bond pricing and the market's information asymmetry. The potential application of the preceding findings will benefit the sustainable development of local government bond market and that of public finance.

## Supporting information

**S1 Data. Orignial data.**
(XLSX)

## Acknowledgments

We appreciate helpful suggestions from Dr. Karren Lee Hwei Khaw (University of Waikato) in the construction of this paper.

## Author Contributions

**Conceptualization:** Changqian Xie, Rubi Ahmad, Eric H. Y. Koh.

**Data curation:** Changqian Xie, Rubi Ahmad, Eric H. Y. Koh.

**Formal analysis:** Changqian Xie.

**Methodology:** Changqian Xie, Rubi Ahmad, Eric H. Y. Koh.

**Resources:** Changqian Xie.

**Supervision:** Rubi Ahmad, Eric H. Y. Koh.

**Validation:** Changqian Xie, Rubi Ahmad, Eric H. Y. Koh.

**Writing – original draft:** Changqian Xie.

**Writing – review & editing:** Changqian Xie, Rubi Ahmad, Eric H. Y. Koh.

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
