## [Decision Letter · Decision Letter 0]

22 Apr 2022

PONE-D-22-08606Does credit rating agency reputation matter in China’s local government bond market?PLOS ONE

Dear Dr. Koh,

Thank you for submitting your manuscript to PLOS ONE. After careful consideration, we feel that it has merit but does not fully meet PLOS ONE’s publication criteria as it currently stands. Therefore, we invite you to submit a revised version of the manuscript that addresses the points raised during the review process.

The submission requires further thoughtful revisions with reference to the econometric framework.

We look forward to receiving your revised manuscript.

Kind regards,

Stefan Cristian Gherghina, PhD. Habil.

Academic Editor

PLOS ONE

Journal Requirements:

Reviewers' comments:

Reviewer's Responses to Questions

**Comments to the Author**

1. Is the manuscript technically sound, and do the data support the conclusions?

Reviewer #1: No

Reviewer #2: Partly

2. Has the statistical analysis been performed appropriately and rigorously? 

Reviewer #1: No

Reviewer #2: No

3. Have the authors made all data underlying the findings in their manuscript fully available?

Reviewer #1: No

Reviewer #2: Yes

4. Is the manuscript presented in an intelligible fashion and written in standard English?

Reviewer #1: Yes

Reviewer #2: Yes

5. Review Comments to the Author

Reviewer #1: The authors examine whether the reputation of credit rating agencies in China shapes the bond yields of local governments. The authors identify an important question to examine. It is also commended that the paper is written with high clarity and coherence. In the spirit of helping the authors improve their paper for plausible publication at a good journal, I offer the following comments and suggestions for the authors.

1. The key assumptions underlying the authors’ analysis are that (i) the purchasers or holders of local governments’ bonds are rational and sophisticated in pricing the bonds of local governments, and that (ii) the bond yield is driven solely by the demand of investors for the bonds and the supply of the high-credit governmental bonds (without interference and dictation by the central government). The authors need to elaborate more on these two assumptions in the hypothesis development. In specific for the first assumption, what kind of investors account for the majority of the purchases and holdings of the local government bonds in China? Is there any evidence or example in practice that these investors really care about, and attend to, the reputation of the credit rating agencies who issue ratings for the local governments’ bonds?

2. Endogeneity is a substantive concern for this study. In particular, those local governments with high creditworthiness and low risk premium in their bonds tend to hire reputable credit rating agencies. This gives rise to the issue of reverse causality in the empirical analysis. The authors conduct the two-stage Heckman regression analysis in an attempt to mitigate the endogeneity concern. However, the instrument the authors choose does not meet the exclusion restriction criteria, because the provincial market shares of the three more reputable rating agencies can also be negatively correlated to the risk premium of the governmental bonds. I advise the authors to use the openness of credit rating industry to foreign credit rating agencies as the exogenous shock to conduct a natural experiment and difference-in-differences regression (DID) analysis. The authors may refer to He et al. (2022) and He et al. (2021), for example, on how to make and implement a reasonable DID research design.

3. The sample period for the baseline regression analysis spans the years 2015-2019, whereas the sample period for the associated moderation analysis covers the years 2009-2014. So there is inconsistency in the sample periods in the empirical tests. I advise the authors to extend the sample period for the baseline regression analysis to 2009-2019. In addition, since the central government unraveled the restrictions on autonomous bond issuance for all local governments in 2015, the reputation effects of credit rating agencies in shaping the governmental bond yield should be stronger because the bond yield would have been driven more by the market forces in such a case. Accordingly, the authors may do a moderation analysis to test this regulatory effect as well. In addition, the authors need to control for local governments’ fiscal transparency in all the regressions, because the fiscal transparency would affect both the bond ratings and bond yield, just as with the case that corporate information transparency would affect credit ratings and bond yield (He 2018). The authors also need to control for GDP and public revenues on top of the GDP growth and public revenues growth.

Reference:

He, G. (2018). Credit ratings and managerial voluntary disclosures. The Financial Review 53(2): 337-378.

He, G., Li, X. & Luo, J. (2021). The impact of Shanghai-Hong Kong stock market connection on corporate innovation: Evidence from mainland China. International Journal of Finance & Economics

He, G., Ren, M.H. & Taffler, R. (2022). Do enhanced derivative disclosures work? An informational perspective. Journal of Futures Markets

Reviewer #2: This paper studies the role of rating agencies' reputation and fiscal transparency of issuers in the risk premium of China's local government bonds. The authors analyse 4846 issues from 2015 to 2019 with a linear regression. They also apply a Heckman's two-stage model to mitigate sample selection bias and endogeneity problems. They find a lower risk premium for bonds rated by more reputable agencies. Less fiscally transparent local governments allow for cost savings by contracting with reputable agencies.

This paper raises an interesting question. However, I have some concerns about the approach, the variables used and the procedures applied that should be resolved before the paper can be published.

Major concerns:

A.I have serious doubts about the variable used to measure the reputation of agencies. This is a central issue in the paper and should be addressed in more detail.

1. It could be that agencies are specialised in certain regions. If this is the case, we may find that the variable captures differences between regions rather than differences between agencies. It is necessary to study the joint distribution of agencies and regions to ensure that this is not the case.

2. Considering that agencies owned by the big three are necessarily those with the best reputations does not seem appropriate. I don't know how these three agencies have penetrated the Chinese market. It is more likely that they have chosen emerging agencies to carry out the operation, rather than the most reputable and established agencies in China. It is possible that their participation has improved these agencies, but they will not necessarily be perceived by the market as more reputable. This may be the result after some time. However, international agencies will enter from 2018 onwards. A much more in-depth study of this question is needed to be able to determine that the agencies in which the big-three agencies participate have indeed been the most reputable since 2015.

3. Other papers directly study the quality of the ratings issued by the agencies to determine which are the best. On other occasions, for example, Bendendo et al. (2018) and Abad et al. (2020) use the bankruptcy of companies with previous high ratings to identify negative shocks to the reputation of the agencies.

4. Some characteristics such as the size of the agency, its age, market share (as mentioned in the paper), its specialisation in public or private debt, etc. can help to identify the most reputable agencies. In any case, the authors should do more and better work to try to convince us that their way of measuring reputation is valid and they should also compare the results of the paper with those obtained with other alternatives.

B. I also have doubts about how fiscal non-transparency is measured. An indicator is used that is constructed with data from 2009 to 2014. I understand that it is a static indicator, which does not vary from year to year between 2015 and the end of the sample. This implies that regions that were not transparent before 2014 will not be transparent afterwards, which does not have to be true. This variable does not measure fiscal transparency, it measures something else. The authors have to include in the model an up-to-date measure of fiscal transparency in order to analyse its effect.

C. Regarding the econometric model, I find that it presents several important problems:

1. It uses the individual issue (the bond) as the unit of observation. However, we have that we can group these bonds by issuer. By not doing so there is a cross-correlation problem in that invalidates the results of the model if it is not explicitly considered.

2. Individual fixed effects are not included, which may also affect the results. At a minimum, fixed effects per issuer should be included.

3. A cluster-robust variance-covariance matrix should be used considering clusters per issuer and year.

D. As a suggestion, other characteristics of the regions should be considered, as the relative size, the relative weight in China's GDP, the relative degree of development, etc. Authors shoud include also an interest rate (1 year Tbill for example) to include time movements in the bond market conditions and/or the slope of the term structure to include expectations about future growth that may covariate with the risk primium.

Minor concerns

- the DR007 variable must be defined

- All definitions in Table 4 must be enhanced

References:

Abad, P., Ferreras, R., & Robles, M. D. (2019). Informational role of rating revisions after reputational events and regulation reforms. International Review of Financial Analysis, 62, 91-103.

Bedendo, M., Cathcart, L., & El-Jahel, L. (2018). Reputational shocks and the information content of credit ratings. Journal of Financial Stability, 34, 44-60.

6. PLOS authors have the option to publish the peer review history of their article (what does this mean?). If published, this will include your full peer review and any attached files.

Reviewer #1: **Yes: **Guanming He

Reviewer #2: No

---

## [Author Response · Author response to Decision Letter 0]

4 Jul 2022

This revised manuscript mainly has the following modifications: (1) The sample period is updated to 2021. Therefore, our research sample extends from 4846 bonds (2015-2019) to 7941 bonds (2015-2021). (2) Owing to the extension of the sample period, we have enough data and conducted the difference-in-differences analysis by using the opening up of China’s rating industry as an exogenous shock. (3) We also add an endogeneity test by using the machine learning method to construct the counterfactual group on top of two usual endogeneity checking methods. (4) We changed the variable of fiscal transparency from static to a dynamic one and included it in our regression model. The moderating effect of fiscal transparency on the reputation effect of rating agencies is also analyzed. (5) Compared to our previous regression model, we improved it by adding more variables about issuers' characteristics and adding the issuer fixed effects and regions fixed effects. 

All the main modifications have been highlighted in the document. The point-by-point responses to the reviewers’ comments are as follow. If there is anything needing further improvement, please notify us without hesitation. We will complete the modification as soon as possible.

Reviewer #1: The authors examine whether the reputation of credit rating agencies in China shapes the bond yields of local governments. The authors identify an important question to examine. It is also commended that the paper is written with high clarity and coherence. In the spirit of helping the authors improve their paper for plausible publication at a good journal, I offer the following comments and suggestions for the authors.

1. The key assumptions underlying the authors’ analysis are that (i) the purchasers or holders of local governments’ bonds are rational and sophisticated in pricing the bonds of local governments, and that (ii) the bond yield is driven solely by the demand of investors for the bonds and the supply of the high-credit governmental bonds (without interference and dictation by the central government). The authors need to elaborate more on these two assumptions in the hypothesis development. In specific for the first assumption, what kind of investors account for the majority of the purchases and holdings of the local government bonds in China? Is there any evidence or example in practice that these investors really care about, and attend to, the reputation of the credit rating agencies who issue ratings for the local governments’ bonds?

Response:

Thank you very much for your suggestion. The supplementary explanation for these assumptions is added and highlighted in the 4th Paragraph of the part of Naive and rational investors assumption (starting from Page 6). 

For the first assumption, the commercial banks and other institutional investors are the major investors in China’s local government bond market, holding 92.16% of the total outstanding amount of local government bonds at the end of 2021 (presented in the footnote 5, Page 6). According to the literature (He et al., 2021, Funaoka & Nishimura, 2019), it can be known that institutional investors are more rational and sophisticated than individual investors and usually make investment decisions based on more comprehensive information. In addition, the content and willingness of provinces’ self-disclosure are relatively limited, which pushes these institutional investors to pay more attention to credit rating agencies' rating jobs. Therefore, we think our hypothesis based on the rational investor assumption is reasonable.

As for your second concern, we have to admit that someone concerned about the potential implicit guarantee from the central government might make this market not solely driven by supply and demand. However, according to the official announcements from the central government (presented in the footnote 6, Page 7), we believe that the willingness of the central government to intervene in the local government bond market is low. At least until now, no default event happened in China’s local government bond market. Therefore, we also deem that it is not necessary for the central government to intervene in this stable situation.

Hope our modification and explanation of this comment can make you satisfied. Thank you very much.

2. Endogeneity is a substantive concern for this study. In particular, those local governments with high creditworthiness and low risk premium in their bonds tend to hire reputable credit rating agencies. This gives rise to the issue of reverse causality in the empirical analysis. The authors conduct the two-stage Heckman regression analysis in an attempt to mitigate the endogeneity concern. However, the instrument the authors choose does not meet the exclusion restriction criteria, because the provincial market shares of the three more reputable rating agencies can also be negatively correlated to the risk premium of the governmental bonds. I advise the authors to use the openness of credit rating industry to foreign credit rating agencies as the exogenous shock to conduct a natural experiment and difference-in-differences regression (DID) analysis. The authors may refer to He et al. (2022) and He et al. (2021), for example, on how to make and implement a reasonable DID research design.

Response:

Thank you very much for your highlighting this concern. According to your these suggestions, we added more explanations and tests to the manuscript.

First, about the two-stage Heckman regression analysis, we are sorry for our poor expression in the previous version. The variable (Reputation_CRA_MarketShare) used in the first stage is the market shares of the three more reputable credit rating agencies in the corporate bond market. Since the corporate bond market and the local bond market are two different markets, we believe this variable is unlikely to be correlated with the error term of local bonds’ risk premiums in the second regression stage, thus meeting the restriction criteria. The explanation in detail is added in the part of Endogeneity Concern: Heckman two-stage model (on Page 18).

Second, your suggestion that we can use the openness of China’s credit rating agency industry as an exogenous shock to conduct the DID analysis inspired us a lot. We design the DID regression based on the method of propensity-score matching (PSM) to group sample, and conduct the necessary test of parallel trends assumption. The results of DID demonstrate that the opening-up policy in 2018, an exogenous shock, has a significant impact on the rating industry in China, thus effectively alleviating the potential endogenous problems in this study. All the results and discussion about the DID are shown in the new part: Endogeneity Concern: Difference-in-differences regression.

Third, in order to further mitigate the potential endogeneity issue (the issue of reverse causality), we also try to use the machine learning method to construct the counterfactual group. The results prove that, with keeping all other conditions unchanged, the only change in the reputation of credit rating agencies of local government bonds (from high reputation in the factual set to low reputation in the counterfactual set), there are significant differences in the risk premiums of the factual and counterfactual sets (the risk premiums of the counterfactual set are higher). Therefore, the causal relationship between Reputation_CRA (the cause) and the risk premium (the effect) is tenable. In this way, the issue of reverse causality in our study can be remediated further. All the results and discussion about this method are shown in the new part: Endogeneity Concern: Machine learning method.

We hope that our modifications to this comment can alleviate the potential endogeneity issue in our study.

3. The sample period for the baseline regression analysis spans the years 2015-2019, whereas the sample period for the associated moderation analysis covers the years 2009-2014. So there is inconsistency in the sample periods in the empirical tests. I advise the authors to extend the sample period for the baseline regression analysis to 2009-2019. In addition, since the central government unraveled the restrictions on autonomous bond issuance for all local governments in 2015, the reputation effects of credit rating agencies in shaping the governmental bond yield should be stronger because the bond yield would have been driven more by the market forces in such a case. Accordingly, the authors may do a moderation analysis to test this regulatory effect as well. In addition, the authors need to control for local governments’ fiscal transparency in all the regressions, because the fiscal transparency would affect both the bond ratings and bond yield, just as with the case that corporate information transparency would affect credit ratings and bond yield (He 2018). The authors also need to control for GDP and public revenues on top of the GDP growth and public revenues growth.

Response:

Thank you very much for your suggestions.

About the study period, we are sorry for our poor expression caused your misunderstanding. We have modified the presented sample period in current Table 2 (on Page 4) to make it consistent with the sample period in our baseline regression. The reason that we use the study period starting from 2015 is that, before 2015, most local government bonds were issued or guaranteed by the central government, which means that these bonds do not fully represent the local governments’ credit risk. The first three stages (2009-2014) are a gradual pilot stage. In addition, there is no compulsory requirement to employ credit rating agencies to do the rating work before 2015. Therefore, for our study, it is not meaningful to include the period from 2009 to 2014, as well as the moderation analysis of the policy in 2015. It is worth noting that we updated our data and extended it to 2015-2021 in this version.

According to your suggestion about adding the fiscal transparency, we have modified the explanation about this variable (Fiscal_Transparency) and revised the description in Table 4 (on Page 11). In addition, according to your suggestion about controlling the GDP and public revenues on top of the GDP growth and public revenues growth, compared to our regressions in the previous version, we include more variables, including the GDP, GDP per capita, GDP growth rate, fixed asset investment growth rate, debt ratio, public revenue per capita, and public revenue growth rate in this revised version. The definition of all variables added is also shown in Table 4 (on Page 11). However, due to the serious multicollinearity between the variable of the GDP and public revenues, we cannot include public revenues in our regression at the same time.

Hope our modification and explanation of this comment can solve your questions. Thank you very much for all your suggestion!

Reviewer #2: This paper studies the role of rating agencies' reputation and fiscal transparency of issuers in the risk premium of China's local government bonds. The authors analyse 4846 issues from 2015 to 2019 with a linear regression. They also apply a Heckman's two-stage model to mitigate sample selection bias and endogeneity problems. They find a lower risk premium for bonds rated by more reputable agencies. Less fiscally transparent local governments allow for cost savings by contracting with reputable agencies.

This paper raises an interesting question. However, I have some concerns about the approach, the variables used and the procedures applied that should be resolved before the paper can be published.

Major concerns:

A.I have serious doubts about the variable used to measure the reputation of agencies. This is a central issue in the paper and should be addressed in more detail.

1. It could be that agencies are specialised in certain regions. If this is the case, we may find that the variable captures differences between regions rather than differences between agencies. It is necessary to study the joint distribution of agencies and regions to ensure that this is not the case.

Response:

Thank you very much for highlighting this concern. To avoid this situation, in this revision, we include the regional fixed effects in our baseline regression on top of issuer fixed effects and year fixed effects, to capture the effects of unobserved regional differences on the dependent variable. The details can be found in current Table 8 (on Page 16), Table 9 (on Page 18), Table 14 (on Page 27), and Table 15 (on Page 28).

In addition, to make it more intuitive, we provide the details about the number of bonds rated by the credit rating agencies with high or low reputation in each region each year in current Table 7 (on Page 15). It can be found that for each year, there is no region where all bonds are only rated by credit rating agencies with high (low) reputation, and there is also no group of credit rating agencies with high (low) reputation only covering a certain region. Therefore, we can ensure that our core explanatory variable (Reputation_CRA) can only capture differences between the high and low reputations of credit rating agencies instead of the differences between regions.

Hope our modification can alleviate your concern about this issue.

2. Considering that agencies owned by the big three are necessarily those with the best reputations does not seem appropriate. I don't know how these three agencies have penetrated the Chinese market. It is more likely that they have chosen emerging agencies to carry out the operation, rather than the most reputable and established agencies in China. It is possible that their participation has improved these agencies, but they will not necessarily be perceived by the market as more reputable. This may be the result after some time. However, international agencies will enter from 2018 onwards. A much more in-depth study of this question is needed to be able to determine that the agencies in which the big-three agencies participate have indeed been the most reputable since 2015.

Response:

Thank you very much for this concern. From three different aspects, we will explain why we use this variable to measure the reputation of credit rating agencies.

First, based on the literature, many recent studies, such as Livingston et al. (2018), Hu et al. 2020, Jiang & Packer (2019), and Xu & Liu (2021) support that this variable is appropriate to measure the reputation of credit rating agencies in China. Han et al. (2012) based on the Japanese bond market, also report that the ratings offered by the more reputable global rating agencies have a stronger effect than those offered by the local ones. Therefore, our selection has the basis of literature.

Second, considering the time since the big three rating agencies (S&P, Moody, and Fitch) come into corporate with their Chinese partners is more than ten years ago, thus, the market participants are already familiar with this. Detailed information about when and how the big three agencies came to China is presented in the footnote 3 (on Page 3). One of our key assumptions is the investors or holders of local government bonds are rational and sophisticated in pricing the bonds of local governments. This assumption can be supported by the fact that the commercial banks and other institutional investors are the major investors in China’s local government bond market, holding 92.16% of the total outstanding amount of local government bonds at the end of 2021 (presented in the footnote 5, Page 6). According to the literature (He et al., 2021, Funaoka & Nishimura, 2019), it can be known that institutional investors are more rational and sophisticated than individual investors and usually make investment decisions based on more comprehensive information. It is reasonable that these rational market participants have already noticed the useful information. Therefore, our selection has the theoretical basis.

Third, according to the other reviewer’s comment, we use the openness of China’s credit rating agency industry in 2018 as an exogenous shock to conduct the difference-in-differences analysis. The results from DID regression are shown in the new part: Endogeneity Concern: Difference-in-differences regression. The results show that, comparing the risk premiums of local government bonds rated by the credit rating agencies with high reputation to those rated by agencies with low reputation, after the foreign agencies entered into China’s bond market, the former is further lower than the latter. This means that the investors or holders recognize that the reputation of credit rating agencies with foreign partners has been further improved. In addition, all the coefficients on Reputation_CRA from our empirical analysis are negative and statistically significant, which means the current group of credit rating agencies with high reputation present the feature of high reputation, to reduce the risk premium. Otherwise, these coefficients should be positive if the current group with high reputation is perceived as low reputation in reality. Therefore, our selection can be proved by the empirical results.

Hope our explanations and modifications to this comment can alleviate your concern.

3. Other papers directly study the quality of the ratings issued by the agencies to determine which are the best. On other occasions, for example, Bendendo et al. (2018) and Abad et al. (2020) use the bankruptcy of companies with previous high ratings to identify negative shocks to the reputation of the agencies.

Response:

Thank you very much for providing two references (Abad et al., 2019; Bendendo et al., 2018) who use the bankruptcy of companies with previous high ratings as negative shocks to study the reputation of rating agencies. However, no default event has happened in China’s local government bond market until now. Therefore, this solution is not very suitable for our study.

4. Some characteristics such as the size of the agency, its age, market share (as mentioned in the paper), its specialisation in public or private debt, etc. can help to identify the most reputable agencies. In any case, the authors should do more and better work to try to convince us that their way of measuring reputation is valid and they should also compare the results of the paper with those obtained with other alternatives.

Response:

Thank you very much for your suggestion. Although the characteristics you mentioned may reflect some features of the reputation, there are some limitations to using these measurements in our study. For example, the data on the size of the agencies are not available for each year because these agencies are not listed in the stock market. If the ages of the agencies are selected, the boundary between high and low reputation is more confusing and subjective. For the market share, some researchers suggest that the market share represents the degree of competition in the Chinese rating industry rather than reputation (Wu & Wang, 2016). In addition, as abovementioned, all the coefficients on Reputation_CRA from our empirical analysis are negative and statistically significant, which means the current group of credit rating agencies with high reputation really present the feature of high reputation, to reduce the risk premium. Therefore, the rationality of our selection can be proved by the empirical results.

In addition, sorry for the misunderstanding caused by our description. In fact, in order to meet the exclusion restriction criteria, the variable of market share (Reputation_CRA_MarketShare) used in the Heckman two-stage model is the market share in the corporate bond market rather than the market share in the local government bond market. Since the corporate bond market and the local bond market are two different markets, we believe this variable is unlikely to be correlated with the error term of local bonds’ risk premiums in the second regression stage, thus meeting the restriction criteria.

Hope our explanations and modifications can alleviate all your concern about our proxy selection to measure the agencies' reputation.

B. I also have doubts about how fiscal non-transparency is measured. An indicator is used that is constructed with data from 2009 to 2014. I understand that it is a static indicator, which does not vary from year to year between 2015 and the end of the sample. This implies that regions that were not transparent before 2014 will not be transparent afterwards, which does not have to be true. This variable does not measure fiscal transparency, it measures something else. The authors have to include in the model an up-to-date measure of fiscal transparency in order to analyse its effect.

Response:

Thank you very much for your suggestion on this issue. Sorry for our careless consideration of this issue in the previous version, we are inspired a lot by your suggestion. We changed the static variable used in the previous version and included the dynamic variable (Fiscal_Transparency) in our regression model. We have modified the explanation about this variable (Fiscal_Transparency) in the manuscript (on Page 10) and also revised the description in Table 4 (on Page 11). Also, we examined its moderating effect on the reputation of rating agencies. The new results and analysis are shown in Table 8, Figure 1, and the corresponding text. All the results support our hypotheses. We hope that our supplementary can solve your question. 

C. Regarding the econometric model, I find that it presents several important problems:

1. It uses the individual issue (the bond) as the unit of observation. However, we have that we can group these bonds by issuer. By not doing so there is a cross-correlation problem in that invalidates the results of the model if it is not explicitly considered.

2. Individual fixed effects are not included, which may also affect the results. At a minimum, fixed effects per issuer should be included.

3. A cluster-robust variance-covariance matrix should be used considering clusters per issuer and year.

Response:

Thank you very much for your suggestion. According to your suggestion, we include the issuer fixed effects, year fixed effects, and region fixed effects in our regression model in this revised version. All the results still support our hypotheses. These new results are shown in Table 8 (on Page 16), Table 11 (on Page 21), Table 12 (on Page 22), Table 14 (on Page 27), and Table 15 (Page 28). We hope that our modifications according to this comment can solve the problems.

D. As a suggestion, other characteristics of the regions should be considered, as the relative size, the relative weight in China's GDP, the relative degree of development, etc. Authors shoud include also an interest rate (1 year Tbill for example) to include time movements in the bond market conditions and/or the slope of the term structure to include expectations about future growth that may covariate with the risk primium.

Response:

Thank you very much for this suggestion. 

Compared to our regressions in the previous version, we include more characteristics of issuers in our regression model, such as the GDP, GDP per capita, GDP growth rate, fixed asset investment growth rate, debt ratio, public revenue per capita, and public revenue growth rate (shown in Table 4 on Page 11) in this revised version. We also try to include the variable of GDP relative weight (defined as the percentage of this issuer’s GDP to total GDP in China) in our regression as you suggested. The regression results that include this variable are shown in the table below.

Table: Regression results including the variable of GDP relative weight

 RiskPremium5 RiskPremium0

Reputation_CRA -6.7432** -12.1864***

 (3.2697) (3.3225)

Fiscal Transparency -2.5594*** -3.0545***

 (0.6579) (0.6687)

RepFis 1.7938** 3.0951***

 (0.8486) (0.8588)

Maturity 0.0596*** 0.0550***

 (0.0129) (0.0117)

Issue Size 0.1989** 0.1624*

 (0.0904) (0.0904)

Issue Frequency 0.2333 -0.2715

 (0.3555) (0.3718)

Bond Type 1.2966*** 1.0931***

 (0.2394) (0.2397)

Sale Methods 25.3714*** 24.6966***

 (0.3392) (0.3605)

GDP Share Year -1.6117* -2.2525**

 (0.9531) (0.9519)

GDP per capita 2.1673** 2.6479***

 (0.9724) (0.9605)

GDP Growth Rate 0.0689 0.1601***

 (0.0604) (0.0611)

FAI Growth Rate 0.0812*** 0.0811***

 (0.0168) (0.0166)

Debt Ratio 0.0103*** 0.0134***

 (0.0036) (0.0036)

Public Revenue per capita 5.5491*** 7.1640***

 (0.9750) (0.9736)

Public Revenue Growth Rate -0.0914*** -0.1201***

 (0.0236) (0.0240)

Tbill 4.2684*** 4.2430***

 (0.3397) (0.3334)

Constant -54.5683*** -67.6652***

 (12.0842) (12.0039)

Year dummies Included Included

Region dummies Included Included

Issuer dummies Included Included

Adjusted R-squared 0.6060 0.6000

No. of observations 7941 7941

The regression results, including the variable of GDP relative weight (GDP Share Year), are consistent with the regression results in Table 8 (on Page 16). However, as the variable of GDP relative weight (GDP Share Year) is highly collinear with the variable of GDP_Year (correlation coefficient is 0.8581), finally, we keep the variable of GDP_Year in our regression model.

According to your suggestion, we add the Tbill to improve our model. The modifications are shown in Table 4 (on Page 11) and also in all regression results tables. 

Minor concerns

- the DR007 variable must be defined

- All definitions in Table 4 must be enhanced

Response:

Thank you very much for this suggestion. 

For DR007, it is an interest rate variable measuring market liquidity, equivalent to The Chinese version of LIBOR interest rate. It is defined as the 7-day repurchase rate of interbank bond market participants with interest rate bonds as pledges. It may reflect the fundamentals of the economy, the trend of seasonal changes, as well as the liquidity risk premium. The detailed definition in English can be found from the website of The People’s Bank of China:

http://www.pbc.gov.cn/en/3688006/3689169/4466612/index.html

Since the DR007 is highly collinear with Tbill, we only kept the Tbill variable in our model and deleted the variable of DR007. Sorry for our previous expression making you confused about this variable. 

On the other aspect, according to your suggestion, we have modified all the definitions in Table 4 (on Page 11). Hope our modifications can make it clearer.

 Thank you very much for all your suggestion! Hope our explanations and modifications can improve our manuscript.

---

## [Decision Letter · Decision Letter 1]

28 Jul 2022

PONE-D-22-08606R1Does credit rating agency reputation matter in China's local government bond market?PLOS ONE

Dear Dr. Koh,

Thank you for submitting your manuscript to PLOS ONE. After careful consideration, we feel that it has merit but does not fully meet PLOS ONE’s publication criteria as it currently stands. Therefore, we invite you to submit a revised version of the manuscript that addresses the points raised during the review process. The submission improved in a proper manner, but a revision towards English language is more than necessary. As well, the author(s) should consider the remaining comments of the second referee.

We look forward to receiving your revised manuscript.

Kind regards,

Stefan Cristian Gherghina, PhD. Habil.

Academic Editor

PLOS ONE

Journal Requirements:

Reviewers' comments:

Reviewer's Responses to Questions

**Comments to the Author**

1. If the authors have adequately addressed your comments raised in a previous round of review and you feel that this manuscript is now acceptable for publication, you may indicate that here to bypass the “Comments to the Author” section, enter your conflict of interest statement in the “Confidential to Editor” section, and submit your "Accept" recommendation.

Reviewer #1: All comments have been addressed

Reviewer #2: (No Response)

2. Is the manuscript technically sound, and do the data support the conclusions?

Reviewer #1: Partly

Reviewer #2: Yes

3. Has the statistical analysis been performed appropriately and rigorously? 

Reviewer #1: Yes

Reviewer #2: Yes

4. Have the authors made all data underlying the findings in their manuscript fully available?

Reviewer #1: No

Reviewer #2: (No Response)

5. Is the manuscript presented in an intelligible fashion and written in standard English?

Reviewer #1: No

Reviewer #2: Yes

6. Review Comments to the Author

Reviewer #1: The paper has improved substantially. I advise the authors to employ a good professional copy-editor to improve the exposition of the paper.

Reviewer #2: The authors have responded adequately almost all my comments. However, I have a concern regarding their response to my comment 3. As the authors are surely aware, there is at least one recent scandal that led to a punishment of the Dagon agency in 2018 (see https://www.bnnbloomberg.ca/china-rating-firm-banned-as-regulators-cite-fake-info-chaos-1.1125717#:~:text=China%20Rating%20Firm%20Banned%20as%20Regulators%20Cite%20Fake,punishment%20ever%20doled%20out%20a%20ratings%20company). This implies a lower reputation of this agency compared to the others. There are also cases of international credit rating misbehaviour outside China, but well known to international markets. All these cases (at least one of them) can be used to test the effect of reputational shocks.

7. PLOS authors have the option to publish the peer review history of their article (what does this mean?). If published, this will include your full peer review and any attached files.

Reviewer #1: **Yes: **Guanming He

Reviewer #2: No

---

## [Author Response · Author response to Decision Letter 1]

4 Aug 2022

The entire manuscript has been native-speaker proofread by a professional company to make sure our expression more accurate and clear. We also checked our reference list, and there is no retracted article. However, six articles were published in the Chinese language, and we have marked in the reference list. The point-by-point responses to the reviewers’ comments are as follow. If there is anything needing further improvement, please notify use without hesitation. We will complete the modification as soon as possible.

Reviewer #1: The paper has improved substantially. I advise the authors to employ a good professional copy-editor to improve the exposition of the paper.

Response:

Thank you very much for your kind suggestions. We sent our manuscript to a professional institution and completed the proofreading. We hope this revised version can meet the requirement of language.

Reviewer #2: The authors have responded adequately almost all my comments. However, I have a concern regarding their response to my comment 3. As the authors are surely aware, there is at least one recent scandal that led to a punishment of the Dagon agency in 2018 (see https://www.bnnbloomberg.ca/china-rating-firm-banned-as-regulators-cite-fake-info-chaos-1.1125717#:~:text=China%20Rating%20Firm%20Banned%20as%20Regulators%20Cite%20Fake,punishment%20ever%20doled%20out%20a%20ratings%20company). This implies a lower reputation of this agency compared to the others. There are also cases of international credit rating misbehaviour outside China, but well known to international markets. All these cases (at least one of them) can be used to test the effect of reputational shocks.

Response:

We apologize for that our unclear explanation made you misunderstand. First, we agree that the punishment of the Dagong agency in 2018 implies a lower reputation for this agency. However, the Dagong agency in our study has been classified in the low reputation group. It is worth noting that we only divided all agencies into two groups (i.e., high and low reputation groups), instead of into more groups with more details, such as the groups with low and lower reputations. Therefore, the influence of this punishment on the reputation of Dagong agency is not suitable to be considered in this study. 

Second, when it comes to other cases of international credit rating misbehavior, all the cases belong to the events that happened in different markets and in different countries. If these cases are included in our manuscript to study their influence, then the analysis will undoubtedly result in some complicated spillover effects. Therefore, previous literature (Livingston et al.,2018; Hu et al.,2020) has not completed a similar analysis, either. 

Lastly, even if there is any event that can influence the reputation of the agencies in China’s local government bond market, it cannot become the criterion to evaluate whether or not these agencies are in the correct groups with high and low-reputation. As you have mentioned, the case of Dagong agency can only make its reputation lower. However, it is not related to its original reputation being high or low. We believe that the results of our baseline regression and robust checks can demonstrate the correction of dividing the current high- and low-reputation groups. In the baseline regression, the results listed in Table 8 show that the coefficients of Reputation_CRA to Risk Premium 5 and Risk Premium 0 are negative and statistically significant. According to the reputation certification hypothesis, only those with high reputation can help reduce bond risk premium. Therefore, the results verify that our selection of high-reputation group enjoys high reputation. If our selection is wrong or opposite (i.e., current high reputation is essentially low reputation, and current low reputation is essentially high reputation), then the coefficients should be positive. Similarly, the result of the second-stage Heckman model (Table 9) shows that the coefficient of Reputation_CRA to Risk Premium 5 is –10.0667. Coefficients of Reputation_CRA to Risk Premium 5 to Risk Premium 0 in the DID regression results in Table 11 are –15.2346 and –14.7596, respectively. In machine learning, differences between the factual set (high reputation) and counterfactual set (low reputation) are –0.103bp and –0.0561bp, respectively. When excluding negative risk premium observations, Table 14 shows that coefficients of Reputation_CRA to Risk Premium 5 to Risk Premium 0 are –5.8774 and –6.4877, respectively. When using the subsample for robustness check, the two coefficients are –9.3151 and –11.7342, as shown in Table 15. All preceding coefficients are negative, demonstrating the correction of our selection. Hence, we think it’s not necessary to include additional analysis on reputation shock.

Thank you very much for your kind comments and hope our explanations can release your concerns.

---

## [Decision Letter · Decision Letter 2]

16 Aug 2022

PONE-D-22-08606R2Does credit rating agency reputation matter in China's local government bond market?PLOS ONE

Dear Dr. Koh,

Thank you for submitting your manuscript to PLOS ONE. After careful consideration, we feel that it has merit but does not fully meet PLOS ONE’s publication criteria as it currently stands. Therefore, we invite you to submit a revised version of the manuscript that addresses the points raised during the review process.

We look forward to receiving your revised manuscript.

Kind regards,

Hung Do

Academic Editor

PLOS ONE

Journal Requirements:

Additional Editor Comments (if provided):

The referees are generally positive with the authors' responses. However, as suggested by the Reviewer #1, there are two remaining issues that the authors need to address more thoroughly: First, the hypothesis H2 needs to be developed in more depth and breadth. Second, the economical significance of the results need to be discussed throughout the paper.

Reviewers' comments:

Reviewer's Responses to Questions

**Comments to the Author**

1. If the authors have adequately addressed your comments raised in a previous round of review and you feel that this manuscript is now acceptable for publication, you may indicate that here to bypass the “Comments to the Author” section, enter your conflict of interest statement in the “Confidential to Editor” section, and submit your "Accept" recommendation.

Reviewer #1: (No Response)

Reviewer #2: All comments have been addressed

2. Is the manuscript technically sound, and do the data support the conclusions?

Reviewer #1: Partly

Reviewer #2: Yes

3. Has the statistical analysis been performed appropriately and rigorously? 

Reviewer #1: Yes

Reviewer #2: Yes

4. Have the authors made all data underlying the findings in their manuscript fully available?

Reviewer #1: Yes

Reviewer #2: Yes

5. Is the manuscript presented in an intelligible fashion and written in standard English?

Reviewer #1: Yes

Reviewer #2: Yes

6. Review Comments to the Author

Reviewer #1: There are two remaining issues for the authors to address: first, the hypothesis H2 needs to be developed in more depth and breadth. Second, the economical significance of the results need to be discussed throughout the paper.

Reviewer #2: I find the authors' response to my comment on reputational shocks satisfactory. However, I encourage the authors to explore this avenue in their future research, as I believe that such an analysis can yield interesting and useful results.

7. PLOS authors have the option to publish the peer review history of their article (what does this mean?). If published, this will include your full peer review and any attached files.

Reviewer #1: **Yes: **Guanming He

Reviewer #2: No

---

## [Author Response · Author response to Decision Letter 2]

30 Aug 2022

This revised manuscript mainly has the following modifications: (1) the second hypothesis development is revised; (2) we added more content to discuss the economic significance of our results throughout the paper. All the main modifications have been highlighted in the document. We also sent the modified content to native-speaker proofreading. The point-by-point responses to the reviewer’s comments are as follow. If anything needs further improvement, please notify us without hesitation. We will complete the modification as soon as possible.

Reviewer #1: There are two remaining issues for the authors to address: first, the hypothesis H2 needs to be developed in more depth and breadth. Second, the economical significance of the results need to be discussed throughout the paper.

Response:

Thank you very much for your kind suggestions. First, we apologize that the explanation and development of hypothesis 2 in the previous version were too brief. In this revised version, we reviewed more previous research and revised the development of hypothesis 2 to meet your requirement (presented on Page 7-Page 8). Second, we added more contents to discuss the economic significance of our results throughout the paper, which are highlighted in the manuscript as follows: 1) Abstract (presented on Page 1); 2) Introduction (presented on Page 2); 3) Baseline regression analysis (presented on Page 15); 4) Endogeneity concern: difference-in-differences regression (presented on Page 21); 5) Conclusion (presented on Page 30). In addition, we sent our manuscript to a professional institution to proofread the revised content again. 

Thank you very much for your kind comments again and hope our modifications can meet your requirements.

Reviewer #2: I find the authors' response to my comment on reputational shocks satisfactory. However, I encourage the authors to explore this avenue in their future research, as I believe that such an analysis can yield interesting and useful results.

Response:

Thank you very much for all your previously valuable suggestions and recognition of our revised work. We would like to set the mentioned analysis as our future research focus according to your suggestion. 

Thank you very much again!

---

## [Decision Letter · Decision Letter 3]

6 Sep 2022

Does credit rating agency reputation matter in China's local government bond market?

PONE-D-22-08606R3

Dear Dr. Koh,

We’re pleased to inform you that your manuscript has been judged scientifically suitable for publication and will be formally accepted for publication once it meets all outstanding technical requirements.

Kind regards,

Hung Do

Academic Editor

PLOS ONE

Additional Editor Comments (optional):

Reviewers' comments:

Reviewer's Responses to Questions

**Comments to the Author**

1. If the authors have adequately addressed your comments raised in a previous round of review and you feel that this manuscript is now acceptable for publication, you may indicate that here to bypass the “Comments to the Author” section, enter your conflict of interest statement in the “Confidential to Editor” section, and submit your "Accept" recommendation.

Reviewer #1: All comments have been addressed

2. Is the manuscript technically sound, and do the data support the conclusions?

Reviewer #1: Yes

3. Has the statistical analysis been performed appropriately and rigorously? 

Reviewer #1: N/A

4. Have the authors made all data underlying the findings in their manuscript fully available?

Reviewer #1: No

5. Is the manuscript presented in an intelligible fashion and written in standard English?

Reviewer #1: Yes

6. Review Comments to the Author

Reviewer #1: The authors have strived to revise the paper to the best possible. I have no further comments on it. Wish the authors enjoy a high impact from the publication.

7. PLOS authors have the option to publish the peer review history of their article (what does this mean?). If published, this will include your full peer review and any attached files.

Reviewer #1: **Yes: **Guanming He

---

## [Editor Report · Acceptance letter]

9 Sep 2022

PONE-D-22-08606R3 

Does credit rating agency reputation matter in China’s local government bond market? 

Dear Dr. Koh:

I'm pleased to inform you that your manuscript has been deemed suitable for publication in PLOS ONE. Congratulations! Your manuscript is now with our production department. 

Kind regards, 

on behalf of

Dr. Hung Do 

Academic Editor

PLOS ONE